# Decoupling Layout from Glyph
# in Online Chinese Handwriting Generation

**Min-Si Ren**[1,2]**, Yan-Ming Zhang**[1,2] *, **Yi Chen**[1,2]
[1]School of Artificial Intelligence, University of Chinese Academy of Sciences,
  Beijing 100049, China
[2]State Key Laboratory of Multimodal Artificial Intelligence Systems (MAIS),
  Institution of Automation Chinese Academy of Sciences,
  Beijing 100190, China
`renminsi17@mails.ucas.ac.cn`
`{ymzhang, yi.chen}@nlpr.ia.ac.cn`

## Abstract

Text plays a crucial role in the transmission of human civilization, and teaching machines to generate online handwritten text in various styles presents an interesting and significant challenge. However, most prior work has concentrated on generating individual Chinese fonts, leaving *complete text line generation largely unexplored*. In this paper, we identify that text lines can naturally be divided into two components: layout and glyphs. Based on this division, we designed a text line layout generator coupled with a diffusion-based stylized font synthesizer to address this challenge hierarchically. More concretely, the layout generator performs in-context-like learning based on the text content and the provided style references to generate positions for each glyph autoregressively. Meanwhile, the font synthesizer which consists of a character embedding dictionary, a multi-scale calligraphy style encoder and a 1D U-Net based diffusion denoiser will generate each font on its position while imitating the calligraphy style extracted from the given style references. Qualitative and quantitative experiments on the CASIA-OLHWDB demonstrate that our method is capable of generating structurally correct and indistinguishable imitation samples. Our source code will be publicly available at: https://github.com/singularityrms/OLHWG .

## 1 Introduction

Deep generative models, such as GANs (Goodfellow et al., 2014), VAEs (Kingma & Welling, 2014), and diffusion models (Sohl-Dickstein et al., 2015; Ho et al., 2021), have demonstrated a formidable ability to generate a wide array of data types. Handwritten data, which encompasses language characters, mathematical symbols, sketches, and diagrams, represents a highly personalized form of data with a wide range of application scenarios. Creating handwriting can provide personalized writing service or be used for data augmentation for document analysis models (Lai et al., 2021; Kang et al., 2021; Xie et al., 2020). In recent years, the emergence of extensive datasets has prompted a notable surge in the application of generative modeling techniques to handwritten data (Xu et al., 2022; Aksan et al., 2018).

Among handwritten data, the generation of Chinese handwritten texts has garnered increasing attention due to their widespread use and the greater challenges they present compared to Latin scripts (Gao et al., 2019; Liu et al., 2022). A crucial aspect of this task is ensuring the integrity and accuracy of the structural composition of the characters. The vast diversity and complex geometric structures of these characters render this task particularly challenging. Another important consideration is how to imitate specific personal writing styles while maintaining structural correctness (Yang et al., 2024; Gan & Wang, 2021; Xie et al., 2021; Kong et al., 2022). Compared to individual characters, text lines serve as more effective communication tools, capable of conveying

---
* Corresponding Author

Figure 1: Handwritten text lines with vastly different styles generated by our methods. It is worth mentioning that the generated online data contains dynamic trajectory information, rather than just static images, enabling more interactive applications. Different colors represent different strokes, showcasing the dynamic process of writing.

more complex information and emotions. However, the majority of prior research has primarily focused on generating individual Chinese fonts, largely overlooking the significantly more challenging task of generating complete text lines.

In this work, we aim to *generate **stylized** and **coherent** online Chinese handwritten text lines based on specified text contents and style reference samples*. A handwritten line of text can be viewed as composed of individual characters and their arrangement. Based on this observation, we propose a hierarchical method that disentangles the layout generation from the writing of individual characters during the generation of full lines. We design a novel text line layout generator to arrange positions for each character based on their categories and given handwriting style references. For font generation, we construct a 1D U-Net (Ronneberger et al., 2015) network and design a multi-scale feature Wang et al. (2023) fusion module to fully mimic the calligraphy characteristics of the given reference sample. We utilize the publicly CASIA-OLHWDB dataset (Liu et al., 2011) for training and evaluating our method, showcasing its effectiveness through a combination of quantitative and qualitative experiments. Figure 1 shows several visualizations of our generated samples.

Our contributions can be summarized as: 1) We propose a hierarchical approach to solve the unexplored online handwritten Chinese text line generation task. 2) We introduce a simple but effective layout generator that can generate character positions based on the text contents and writing style through in-context-like learning. 3) We construct a 1D U-Net-based network for font generation with a multi-scale contrastive-learning-based style encoder to improve the ability of calligraphy style imitation.

## 2 RELATED WORK

### 2.1 ONLINE HANDWRITTEN DATA GENERATION

The most commonly adopted method (Graves, 2013) for generating online handwritten data involves combining neural network models for sequence data processing with a mixture of Gaussian distributions to model the movement information of the pen. Following this work, SketchRNN (Ha & Eck, 2018) adopts RNN as the encoder and decoder of VAE, conducting the task of unconditional sequence data generation. COSE (Aksan et al., 2020) treats drawings as a collection of strokes and projects variable-length strokes into a latent space of fixed dimension and uses MLPs to generate one single stroke based on its latent encoding autoregressively. Subsequently, there has been a growing emphasis in academic research on conditional generation, such as producing handwriting with specific contents or diverse writing styles (Aksan et al., 2018; Kotani et al., 2020; Tolosana et al., 2021). They extract calligraphic styles from reference samples and combine them with specific textual content to achieve controllable style English handwriting synthesis.

In contrast to alphabetic languages, Chinese encompasses a significantly larger character set and Chinese characters exhibit more intricate shapes and topological structures (Xu et al., 2009; Lin et al., 2015; Lian et al., 2018; Radford et al., 2016; Zhu et al., 2017). For this reason, *most methods designed for handwritten English generation do not work well when applied directly to Chinese*.

As for the field of online Chinese character generation, both LSTM and GRU models are employed simultaneously in (Zhang et al., 2017) to successfully generate readable Chinese characters of specific classes at the first time. FontRNN (Tang et al., 2019) focuses on stylized font generation, which utilizes a transfer-learning strategy to generate stylized Chinese characters. However, each trained model can only synthesize the same style as the train set. By adding a style encoder branch to the generator, DeepImitator (Zhao et al., 2020) can generate characters of any style given a few reference samples. They utilize a CNN as the calligraphy style encoder to extract style codes from offline images and a GRU as the generator to synthesize the online trajectories. Similarly, WriteLikeU (Tang & Lian, 2021), DiffWriter (Ren et al., 2023), and SDT (Dai et al., 2023) are also capable of handling this functionality.

## 2.2 DIFFUSION MODEL FOR SEQUENCE DATA

Diffusion model, first proposed in (Sohl-Dickstein et al., 2015) exhibits a remarkably robust capability in learning the data distribution and generating impressively high quality and diverse images in recent years (Luo, 2022; Ho et al., 2021; Dhariwal & Nichol, 2021; Song et al., 2020; Rombach et al., 2021). In terms of sequence data generation, NaturalSpeech2 (Shen et al., 2024) employs the latent diffusion framework to synthesize speech content with specific contents by imitating various tones and speaking styles. CHIRODIFF (Das et al., 2023) first utilizes diffusion for unconditional chirographic data generation and later DiffWriter (Ren et al., 2023) adopts diffusion models for online Chinese characters generation. *However, previous methods only explored how to apply the diffusion process to online handwritten data, lacking fine control over writing details.*

## 3 METHODS

### 3.1 PRELIMINARY

**Online Handwritten Data.** In general, online handwriting is a kind of *time series data*, which is composed of a series of trajectory points. Each point ($[h, v, s] \in \mathbb{R}^3$) contains the *horizontal movement*, *vertical movement* and *the state information* of the pen. We use $s = 1$ to represent the pen touching state and $s = -1$ to represent the pen lifting state, making it convenient to distinguish the two states by simply adopting dichotomy with threshold 0 after the addition of zero-mean standard Gaussian noise to the data.

**Data Notation.** A handwritten font that consists of $n$ points can be represented as $\mathbf{x} \in \mathbb{R}^{n \times 3}$. Its corresponding category is represented as a one-hot vector $\mathbf{c} \in \mathbb{R}^K$, where $K$ is the number of character categories. A handwritten text line can be represented as $\mathbf{X} = [\mathbf{x}^1, \cdots, \mathbf{x}^m] \in \mathbb{R}^{N \times 3}$, where $\mathbf{x}^i \in \mathbb{R}^{n^i \times 3}$ denotes the $i$-th character, $m$ denotes the number of characters, and $N$ denotes the number of points with $N = \sum_{i=1}^m n^i$. Its text content can be represented as $\mathbf{C} = [\mathbf{c}^1, ..., \mathbf{c}^m] \in \mathbb{R}^{m \times K}$.

**Problem Statement.** Given the text line content $\mathbf{C}$ and style reference samples $\mathbf{X}_{ref}$ (preferably a coherent handwritten text line) from a given author, our objective is to generate handwritten text lines with the specified content while imitating the author's writing habits. The conditional probability can be written as $p(\mathbf{X}|\mathbf{C}, \mathbf{X}_{ref})$. Unlike English, Chinese has a large number of character categories, each with its specific stroke order and geometric structure. The combinations to form a text line are endless. Therefore, the **most critical challenge** in generating handwritten Chinese text lines lies in two aspects: *1) Ensuring the correctness of the structure of each character while maintaining consistency with the chosen writing style. 2) Arranging the relative positions of different characters, especially between Chinese characters and punctuation marks, to achieve a natural and fluent layout for the entire handwritten text line.*

### 3.2 ANALYSIS AND OVERVIEW

Directly modeling $p(\mathbf{X}|\mathbf{C}, \mathbf{X}_{ref})$ is overwhelmingly challenging since the positions of glyphs are unknown. In this work, we treat the text line layout as latent variables in the text line generation and instead model the joint distribution of the text line and its layout: $p(\mathbf{X}, \text{Layout}_X|\mathbf{C}, \mathbf{X}_{ref})$.

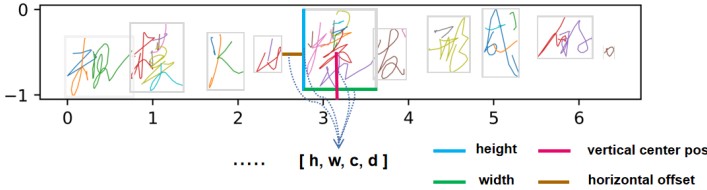

Figure 2: The illustration of character bounding box, consists of height, width, vertical center position and horizontal offset relative to the previous character.

In detail, we define the layout of a handwritten text line as the positions and sizes of all characters in the line. We calculate the [height, width, vertical center position and horizontal offset relative to the previous character] for each character as its bounding box, shown in Figure 2. In this way, the layout of $\mathbf{X}$ can be represented as: $\text{Layout}_X = [\text{Box}^1, \cdots, \text{Box}^m] \in \mathbb{R}^{m \times 4}$.

During the writing process, human beings typically take neighboring glyph positions into account for the current character position to ensure the continuity of the written text lines. However, when the positional information of each character is provided, the writing process of each character is relatively independent. Similarly, we assume that when the bounding box of a character is given, the generation of the character is independent of other characters. By this assumption, we can factorize $p(\mathbf{X}, \text{Layout}_X | \mathbf{C}, \mathbf{X}_{ref})$ as:

$$p(\mathbf{X}, \text{Layout}_X | \mathbf{C}, \mathbf{X}_{ref}) = p(\text{Layout}_X | \mathbf{C}, \mathbf{X}_{ref}) p(\mathbf{X} | \mathbf{C}, \mathbf{X}_{ref}, \text{Layout}_X)$$

$$= p(\text{Layout}_X | \mathbf{C}, \mathbf{X}_{ref}) \prod_{i=1}^{m} p(\mathbf{x}^i | \mathbf{c}^i, \mathbf{X}_{ref}, \text{Box}^i). \tag{1}$$

Equation 1 inspires us to decouple the complete generation process into two components: **layout generation** $p(\text{Layout}_X | \mathbf{C}, \mathbf{X}_{ref})$ and **individual font synthesis** $p(\mathbf{x}^i | \mathbf{c}^i, \mathbf{X}_{ref}, \text{Box}^i)$. Correspondingly, our proposed model consists of two parts: *a text line layout generator* and *a stylized font synthesizer*, shown in Figure 3.

### 3.3 IN-CONTEXT LAYOUT GENERATOR

**Modeling The Human Writing Process.** Considering the consistent and coherent style of the entire text layout when people write a complete text line, it is natural and reasonable to use a certain length of the prefix of the text line as context to predict the subsequent layout. Based on this observation, we decompose the generation of layout in an autoregressive manner:

$$p(\text{Layout}_X | \mathbf{C}, \mathbf{X}_{ref}) = p([\text{Box}^1, \cdots, \text{Box}^m] | \mathbf{C}, \mathbf{X}_{ref})$$

$$= \prod_{i=1}^{m} p(\text{Box}^i | [\text{Box}^1, \cdots, \text{Box}^{i-1}], \mathbf{c}^i, \mathbf{X}_{ref}), \tag{2}$$

which is akin to the human writing process. Denote the layout generator model as $\mathcal{G}$, which is an LSTM network in our work. The generation of the $i$-th character box can be represented as:

$$[\text{Box}^i, \mathbf{h}^i] = \mathcal{G}(\text{Box}^{i-1}, \mathbf{c}^i, \mathbf{h}^{i-1}), \tag{3}$$

where $\mathbf{h}$ represents the hidden state of LSTM. It is worth noting that *different glyphs, punctuation marks, and numbers generally have markedly different shapes*. Therefore, it is crucial to consider the character categories during the generation.

**Training Objective.** During the training phase, we employ the teacher-forcing technique and use the $\ell_1$ distance between ground truth and generated layout as the loss function. We empirically illustrate that this training method encourages the model to generate layouts that are consistent throughout the whole text line.

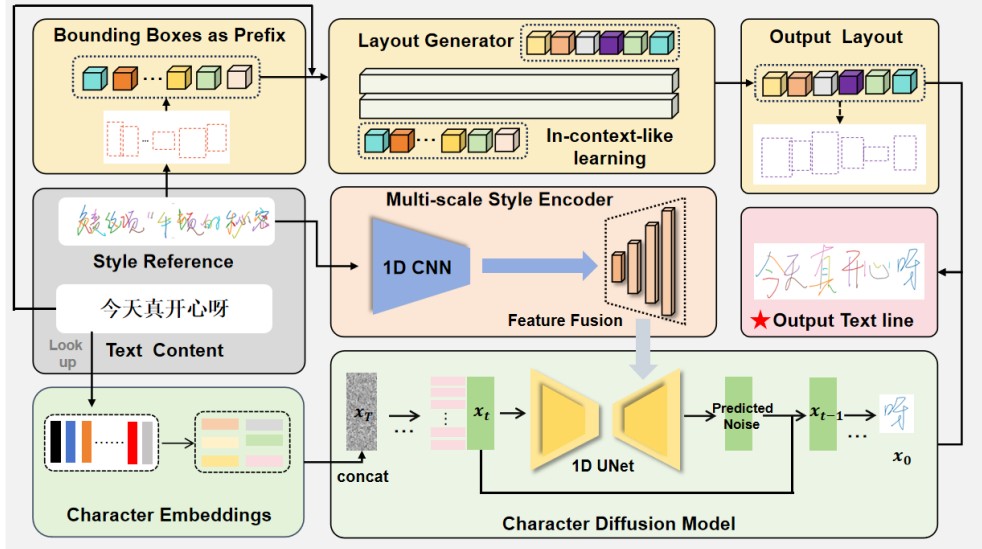

Figure 3: Overview of the proposed method, which consists of a layout generator and a font synthesizer. Given the text content and style references, the two modules operate simultaneously: the layout generator will arrange the bounding box of each character based on the overall style of the reference, while the font synthesizer will imitate the calligraphic style of the references to produce the corresponding handwritten fonts.

**In-Context Generation.** In order to mimic the layout style of the given reference sample, we *use the style reference samples as a prompt context*, inputting their true bounding boxes as a prefix instead of the generator $\mathcal{G}$'s outputs. The outputs corresponding to the prefix time steps are inconsequential, but the hidden state of the LSTM network implicitly contains information about the layout style. It will plan the subsequent layout according to the writing style of the prefix context.

We experimentally show that this approach effectively possesses the capability for in-context-like learning in this task, and therefore can imitate general writing styles unseen in the training set. Additionally, if no coherent text line is provided as the style reference sample, the model will perform unconditional layout planning with a prefix length of 0.

### 3.4 STYLIZED DIFFUSION CHARACTER SYNTHESIZER

We adopt the conditional diffusion model as the generator for individual characters. However, previous diffusion-based methods for online handwritten data have neglected the multi-scale features of calligraphy style. To address this issue, we design a 1D U-Net network as the denoiser, and combine it with a character embedding dictionary and a multi-scale calligraphy style encoder to generate characters with specific categories and calligraphy styles.

#### 3.4.1 MODEL ARCHITECTURE

**Character Embeddings.** Inspired by (Zhang et al., 2017), a dictionary is constructed to encode the structural information of characters: $\mathbf{E} \in \mathbb{R}^{K \times N_z}$, where $K$ represents the number of character categories and $N_z$ represents the code dimension. Each row in matrix E encodes the structure of one category. To generate a character of the $k$-th category, we first query the embedding dictionary and get the $k$-th row of matrix $\mathbf{E}[k] \in \mathbb{R}^{N_z}$ as the content code. Supposing the shape of noisy data $\mathbf{x}_t$ is $(n, 3)$, we duplicate the content code as well as sinusoidal position encoding of time $\text{emb}_t \in \mathbb{R}^{N_t}$ for $n$ times and concatenate them with the noisy data $\mathbf{x}_t$ to obtain the input $\mathbf{x}_{in} = [\mathbf{x}_t, \text{emb}_t, \mathbf{E}[k]] \in \mathbb{R}^{n \times (N_z + N_t + 3)}$ of the denoiser.

**Multi-Scale Style Encoder.** Compared to the global layout style, the calligraphy styles of individual characters are often reflected in various details, such as overall neatness, stroke length, and curvature.

Considering that these characteristics are all at different scales, we construct a 1D convolutional style encoder that employs continuous downsampling layers to extract calligraphy features at different levels and a 1D U-Net denoiser to leverage these features.

**U-Net Denoiser.** The U-Net network *shares the same number of down-sampling layers and strides as the style encoder*. We set the number of down-sampling blocks as 3 in this work. This alignment ensures that during the decoding process, each layer's features in the U-Net can correspond to features of the same scale in the style encoder. These features are fused using the QKV cross-attention mechanism (Rombach et al., 2021), thereby incorporating multi-scale style information into the denoiser. The architectural diagram and further details can be found in Appendix A.2.2.

### 3.4.2 TRAINING OBJECTIVE

Denote the font synthesizer as $\mathcal{E}$, the loss function consists of two parts: a multi-scale style contrastive learning loss $\ell_c$ as well as a diffusion process reconstruction loss $\ell_r$:

$$\ell_{\mathcal{E}} = \ell_c + \ell_r. \tag{4}$$

We detail the two learning objectives as follows.

**Multi-Scale Style Contrastive Learning Loss.** To incentivize the style encoder to extract discriminative calligraphic style features from different authors across diverse scales, we perform contrastive learning on distinct feature layers respectively. More concretely, suppose a mini-batch $\{\mathbf{x}_{w_1}, \cdots, \mathbf{x}_{w_b}\}$ contains $b$ characters written by $b$ different authors $W = \{w_1, \cdots, w_b\}$ and let $A(w_i) = W \backslash \{w_i\}$. Recalling section 3.4.1, the style encoder SENC contains three downsampling blocks corresponding to three different scales of features for one character: $\text{SENC}(\mathbf{x}_{w_i}) = \{\boldsymbol{f}_{w_i}^1, \boldsymbol{f}_{w_i}^2, \boldsymbol{f}_{w_i}^3\}$.

For each scale features, take $\boldsymbol{f}_{w_i}^3 \in \mathbb{R}^{L \times d}$ as an sample, where $L$ represents the length of the feature sequence and $d$ indicates the dimension of each feature vector. We randomly select two non-overlapping segments $\boldsymbol{e}_{w_i}^3, \boldsymbol{e}_{w_i}^{3+}$ from $\boldsymbol{f}_{w_i}^3$ ($\boldsymbol{e} \in \mathbb{R}^{l \times d}$ from the total length $L$), serving as a positive pair from author $w_i$. The contrastive loss for $f^3$ is formulated as:

$$\ell_c^3 = -\frac{1}{n} \sum_{k=1}^{n} \log \frac{\exp\left(s(\boldsymbol{e}_{w_k}^3, \boldsymbol{e}_{w_k}^{3+})/\tau\right)}{\sum_{w_i \in A(w_k)} \exp\left(s(\boldsymbol{e}_{w_k}^3, \boldsymbol{e}_{w_i}^{3+})/\tau\right)}, \tag{5}$$

where $s(\boldsymbol{e}_{w_k}^3, \boldsymbol{e}_{w_k}^{3+}) = g^3(\boldsymbol{e}_{w_k}^3)^T g^3(\boldsymbol{e}_{w_k}^{3+})$, $\tau$ is a temperature parameter, $g^3$ is a linear projection. Similarly, we can obtain $\ell_c^1$ and $\ell_c^2$. The overall multi-scale contrastive loss is formulated as:

$$\ell_c = \lambda_1 \ell_c^1 + \lambda_2 \ell_c^2 + \lambda_3 \ell_c^3, \tag{6}$$

where $\lambda_1, \lambda_2, \lambda_3$ are empirical weights. We set $\lambda_1 = 0.01, \lambda_2 = \lambda_3 = 0.1$ in this work.

**Diffusion Reconstruction Loss and Generation Process.** Given a target character $\mathbf{x}$ and style reference samples $\mathbf{X}_{ref}$ which is a piece of handwritten text line written by the same writer. Denoting the style encoder as SENC, and the character embedding dictionary as E, the diffusion reconstruction loss is calculated in Algorithm 1. $T, \{\alpha_t\}, \overline{\alpha}_t$ are hyperparameters of diffusion models, which are introduced in Appendix A.1. Correspondingly, given the character category $k$ and style reference samples $\mathbf{X}_{ref}$, we randomly sample standard Gaussian noise as $\mathbf{x}_T \in \mathbb{R}^{(n \times 3)}$, where $n$ is the max length, the generation procedure is summarized in Algorithm 2.

## 4 EXPERIMENTS

In the experimental section, we assess our approach in three areas: **capability of calligraphy style imitation**, **capability of layout style imitation**, and **content readability**, utilizing both qualitative and quantitative experiments as followings:

### 4.1 DATASET

We use the CASIA Online Chinese Handwriting Databases (Liu et al., 2011) to train and test our model. For single character generation, following previous work, the CASIA-OLHWDB (1.0-1.2) is

---

**Algorithm 1** Diffusion Reconstruction Loss $\ell_r$

1: **procedure** TRAIN( $\mathbf{x}, \mathbf{X}_{ref}, k, T, \{\alpha_t\}$ )
2:
3:      Sample $t \in (1, T), \epsilon \in \mathcal{N}(0, 1)$
4:      $\mathbf{x}_t = \sqrt{\overline{\alpha}_t}\mathbf{x}_0 + \sqrt{1 - \overline{\alpha}_t}\epsilon$
5:      $\mathbf{x}_{in} = \text{concat}(\mathbf{x}_t, \text{emb}_t, \text{E}[k])$
6:      $f_{style} = \text{SENC}(\mathbf{X}_{ref})$
7:      $\epsilon_\theta = \mathcal{E}(\mathbf{x}_{in}, f_{style})$
8:      $\ell_r(\theta) = ||\epsilon - \epsilon_\theta||^2$
9:      **Output:** $\ell_r$

10: **end procedure**

---

**Algorithm 2** Font Generation Process

1: **procedure** GEN($\mathbf{x}_T, k, \mathbf{X}_{ref}, T, \{\alpha_t\}$)
2:      $f_{style} = \text{SENC}(\mathbf{X}_{ref})$
3:      **for** $t = T, T - 1, ..., 1$ **do**
4:          $\mathbf{x}_{in} = \text{concat}(\mathbf{x}_t, \text{emb}_t, \text{E}[k])$
5:          $\epsilon_\theta = \mathcal{E}(\mathbf{x}_{in}, f_{style})$
6:          $\mu_\theta(\mathbf{x}_t) = \frac{1}{\sqrt{\alpha_t}}\mathbf{x}_t - \frac{1-\alpha_t}{\sqrt{1-\overline{\alpha}_t}\sqrt{\alpha_t}}\epsilon_\theta$
7:          $\sigma_t^2 = \frac{(1-\alpha_t)(1-\overline{\alpha}_{t-1})}{1-\overline{\alpha}_t}$
8:          $\mathbf{x}_{t-1} = \mu_\theta(\mathbf{x}_t) + \sigma_t z, z \sim \mathcal{N}(0, I)$
9:      **end for**
10:      **Output:** $\mathbf{x}_0$
11: **end procedure**

---

adopted as the training set, which contains about 3.7 million online Chinese handwritten characters produced by 1,020 writers. The ICDAR-2013 competition database (Yin et al., 2013b) is adopted as the test set, which contains 60 writers, with each contributing the 3,755 most frequently used characters set of GB2312-80. For layout and text line generation, we adopt CASIA-OLHWDB (2.0-2.2) which consists of approximately 52,000 text lines written by 1,200 authors, totaling 1.3 million characters. We take 1,000 writers as the training set and the left 200 writers as the test set. Detailed descriptions of data preprocessing, model architecture, and training process are provided in Appendix A.2.

## 4.2 FONT GENERATION

**Evaluation Metrics.** To assess the model's ability to imitate calligraphy style and generate accurate character structures, we adopt three core indicators DTW (Normalized Dynamic Time Warping Distance), Content Score, and Style Score following previous work (see Appendix A.2 for details). For each writing style in the test set, we randomly generate 100 characters, totaling 6,000 characters, to measure the performance.

**Baselines and Results.** There are mainly two types of online handwritten Chinese character generation approaches: the first type comprises purely data-driven methods that learn the structure of characters and calligraphy style only from the training set. The second type requires a standard printed font image as an auxiliary input, and train models to transform the standard fonts into characters with a specific writing style. The advantage of the second method is that it is easier to generate correct structures, but it cannot generate characters without corresponding standard glyphs and the computational cost will be heavier. As shown in Table 1, being a method of the first type, our model achieves state-of-the-art performance in all pure data-driven approaches and is comparable to state-of-the-art style transfer-based methods. The qualitative comparison experiment is shown in Appendix A.2.5.

**Effects of Contrastive Learning.** In this experiment, we randomly select 5 writers in the test set, 150 characters for each writer. We first use the style encoder to extract their features and visualize them using the $t$-SNE method (Van der Maaten & Hinton, 2008). As shown in Figure 4, the right side was obtained through our fully trained style encoder, while the left side does not adopt contrastive learning loss. In the right figure, features from the same author usually exhibit a *clear clustering trend*, while in the left figure, the distribution is more *random and scattered*. It illustrates that through contrastive learning, the calligraphy style encoder can extract discriminative features effectively.

Table 1: Evaluation of Character Structure Accuracy and Calligraphy Style Imitation.

| Methods | Purely data-driven | | | Style transfer | | |
|---|---|---|---|---|---|---|
| | DeepImitator | DiffWriter | **Ours** | FontRNN | WriteLikeU | SDT |
| DTW ($\downarrow$) | 1.062 | 0.997 | **0.932** | 1.045 | 0.982 | 0.880 |
| Content Score ($\uparrow$) | 0.834 | 0.887 | **0.891** | 0.875 | 0.938 | 0.970 |
| Style Score ($\uparrow$) | 0.432 | 0.481 | **0.918** | 0.461 | 0.711 | 0.945 |

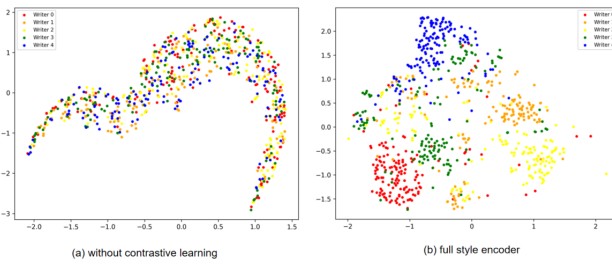

Figure 5: The ablation of multi-scale contrastive learning.

| $\lambda_1$ | $\lambda_2$ | $\lambda_3$ | Style Score | Content Score |
|------|------|------|-------------|---------------|
| 0.0 | 0.0 | 0.0 | 0.875 | 0.893 |
| 0.0 | 0.0 | 0.1 | 0.900 | 0.887 |
| 0.01 | 0.1 | 0.1 | 0.918 | 0.891 |

Figure 4: $t$-SNE visualization of the font style features. Different colors represent different writers.

**Different Weights for Contrastive Learning.** In Table 5, $\lambda_1, \lambda_2, \lambda_3$ are empirical weights in equation 6. A weight of 0 indicates that the loss is not used in this feature layer. We find that while the Content Score is basically unaffected, the Style Score experiences a certain improvement, particularly with the implementation of multi-scale contrastive learning.

### 4.3 FULL LINE GENERATION

#### 4.3.1 EVALUATION METRICS

**Layout Assessment.** The core of our method lies in the in-context layout generation. To better demonstrate the effectiveness of our method, we adopt several *binary geometric features* from traditional OCR (Zhou et al., 2007; Yin et al., 2013a) that can effectively reflect the characteristics of character spatial relations within text lines, shown in Table 2. We calculate the difference in these features between the generated samples and the real samples as quantitative evaluation indicators, denoted as $\nabla_1$ to $\nabla_8$.

Table 2: Binary geometric features for layout generation evaluation, which is calculated between every two adjacent characters.

| No. | Binary geometric features |
|-----|---------------------------|
| 1 | Mean vertical distance between geometric centers |
| 2 | Mean horizontal distance between geometric centers |
| 3 | Mean distance between upper bounds |
| 4 | Mean distance between lower bounds |
| 5 | Mean distance between left bounds |
| 6 | Mean distance between right bounds |
| 7 | Mean ratio of heights of bounding boxes |
| 8 | Mean ratio of widths of bounding boxes |

**Readability Assessment.** To assess the structural correctness performance, *AR (Accurate Rate) and CR (Correct Rate)* are two widely used metrics for Chinese string recognition (Su et al., 2009; Wang et al., 2011). We adopt the methods proposed in (Chen et al., 2023) as the text line recognizer, which can get 0.962 AR and 0.958 CR on the test set. We report the recognition metrics of the recognizer on our synthesized text lines, where *higher values indicate more complete and accurate geometric structure*. More details are introduced in Appendix A.2.4.

**Baselines.** 1) In the field of text line recognition, several methods (Peng et al., 2019; Yu et al., 2024) employs synthesized samples to augment the dataset for training. In these synthesized samples, the bounding box (introduced in Section 3.2) of each glyph category is statistically modeled as a Gaussian distribution in the training set. When generating, the bounding box for each character is sampled based on these Gaussian distributions. We denote this layout generation method as *'Gaussian'*. 2) Furthermore, as mentioned before, if no reference samples are provided, our layout generator model can perform unconditional generation, denoted as *'Unconditional'*. We combine the two layout generation methods with our font synthesizer and compare them with our proposed approach *quantitatively and qualitatively*. The default reference text line length is set as 10.

#### 4.3.2 QUANTITATIVE RESULTS

As shown in Table 3, we find that: 1) the generated text lines all exhibit AR and CR values exceeding 0.85, demonstrating the complete and correct structures of each character. 2) The layouts generated by the in-context method are closer to real samples in nearly all binary geometric features, which highlights the effectiveness of our proposed layout generative method.

Table 3: Evaluation of Handwritten Text Line Layout Imitation and Readability.

| Baselines | $\nabla_1(\downarrow)$ | $\nabla_2(\downarrow)$ | $\nabla_3(\downarrow)$ | $\nabla_4(\downarrow)$ | $\nabla_5(\downarrow)$ | $\nabla_6(\downarrow)$ | $\nabla_7(\downarrow)$ | $\nabla_8(\downarrow)$ | AR | CR |
|---|---|---|---|---|---|---|---|---|---|---|
| Gaussian | 0.074 | 0.151 | 0.092 | 0.092 | 0.163 | 0.160 | 0.859 | 1.081 | 0.851 | 0.855 |
| Unconditional | 0.048 | 0.132 | 0.034 | 0.063 | 0.160 | 0.139 | 0.366 | 0.408 | 0.852 | 0.856 |
| In-Context | 0.046 | 0.122 | 0.062 | 0.058 | 0.129 | 0.129 | 0.364 | 0.419 | 0.852 | 0.857 |

We further investigate the impact of *varying reference sample lengths* on the performance of layout generator as shown in Figure 6. Intuitively, with increasing length, most metrics tend to improve. Compared to the vertical binary geometric features ($\nabla_1, \nabla_3, \nabla_4, \nabla_7$), the horizontal features ($\nabla_2, \nabla_5, \nabla_6, \nabla_8$) exhibit a larger deviation from real samples, indicating that learning the spacing size is more challenging when imitating the layout style of text lines. Furthermore, with respect to these horizontal features, the disparity between the generated samples and the authentic samples is markedly decreased as the length of the reference sample extends.

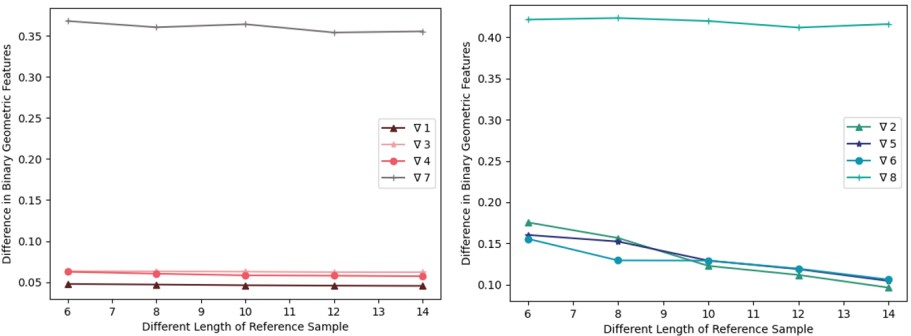

Figure 6: The difference between the mean binary geometric features of generated samples and real samples under different reference sample lengths.

### 4.3.3 QUALITATIVE RESULTS.

**Layout Case Study.** We specifically select samples with unique layout styles (upward slant) in the test set. Figure 7 shows that although the unconditional layout generation method can produce structurally reasonable layouts, it completely neglects the layout style of the handwritten text line. In comparison, our method *effectively mimics characteristics such as glyph spacing and the slant trend of handwriting*, verifying the necessity of our layout generator. Additional experimental results can be found in Appendix A.4.

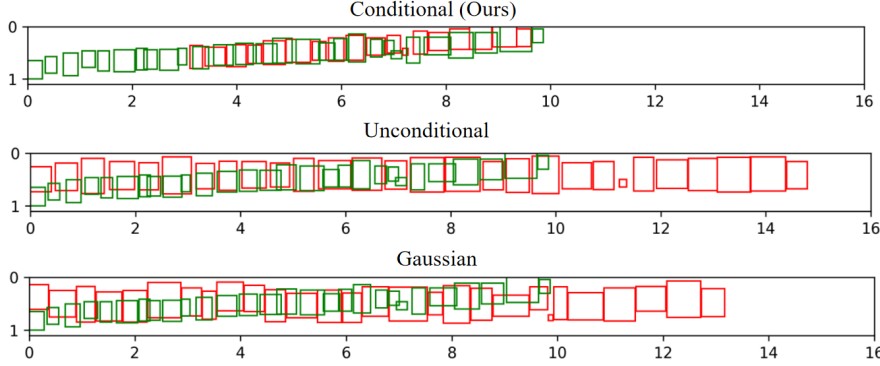

Figure 7: Visualization of character bounding boxes generated by different methods, where green represents the ground truth. For our method, the bounding boxes of the first ten characters are treated as the context prefix.

**Subjective Visualization Experiment.** We construct a visual subjective test as shown in Figure 8. Fifteen pairs of similar samples are created, and 20 participants are asked to determine whether lines in one block were written by the same person. Over 85% of the responses are "yes," indicating that the distinction between real and synthesized samples is difficult to perceive. It can be observed that our model adeptly imitates the target samples, *capturing both the overall layout style, such as spacing and slant, as well as the calligraphic style of individual characters*.

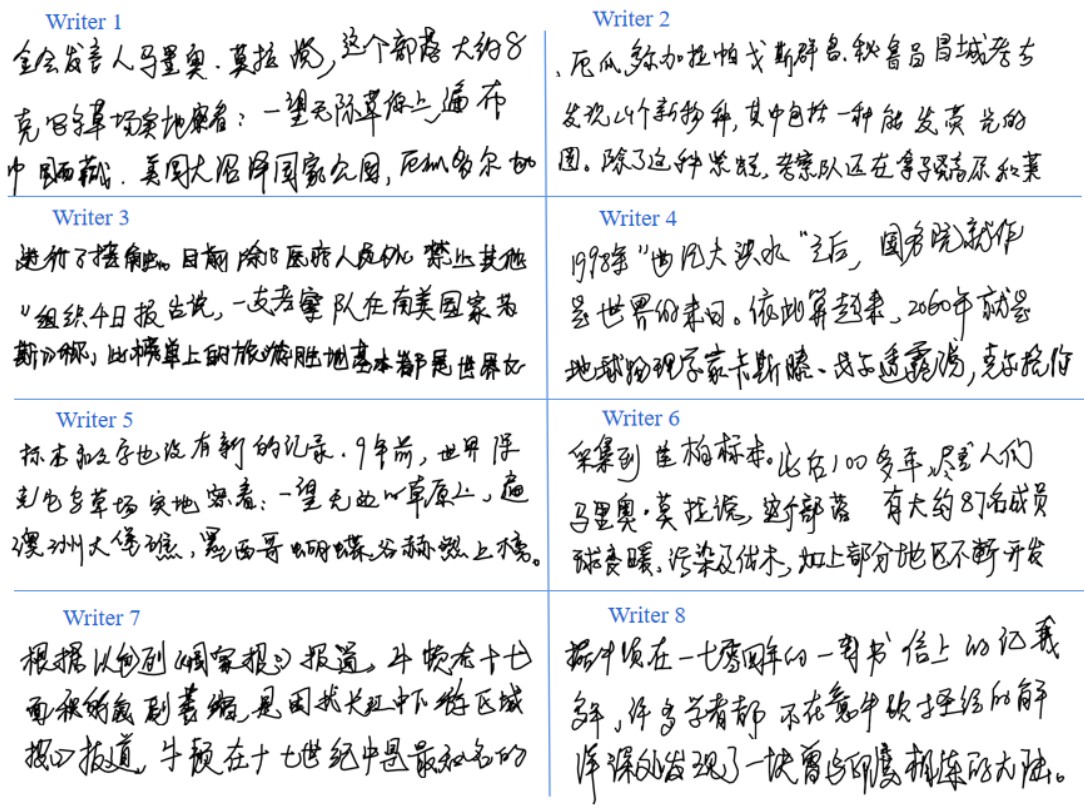

Figure 8: Examples of subjective experiments. For each author block, one row contains real samples and two rows contain synthesized samples. Participants must judge whether the samples were written by the same person.

## 5 CONCLUSIONS AND LIMITATIONS

In this paper, we tackle the stylized online handwritten Chinese text line generation task hierarchically, which is almost unexplored. Our model consists of a novel text line layout generator and a stylized diffusion-based character synthesizer. The text line layout generator can arrange glyph positions based on the text contents and writing habits of the given reference samples while the stylized diffusion-based character synthesizer can generate characters with specific categories and calligraphy styles. However, due to the fact that each character is generated independently, our approach encounters difficulties in mimicking styles with extensive cursive connections between characters. An end-to-end text line generation method might be required to address this issue, as discussed in Appendix A.3. Furthermore, whether synthesized samples can serve as augmentation data to assist in achieving better results for the recognizer is an open question worth exploring. Last but not least, how to generate more complex handwritten data, such as mathematical formulas based on the hierarchical approach is also an intriguing topic. We consider these as our future work.

**Acknowledgments:** This work is supported by the Natural Science Foundation of China (NSFC) Grant No. 62436009, No. 62276258, and U23B2029.

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

# A APPENDIX

## A.1 DENOISING DIFFUSION PROBABILISTIC MODELS

In this paper, we adopt the Denoising Diffusion Probabilistic Model (DDPM), which is a generative model that operates by iteratively applying a denoising process to noise-corrupted data. This process, known as the reverse denoising process aims to gradually refine the noisy input towards generating realistic samples. DDPM learns to model the conditional distribution of clean data given noisy inputs, which is derived from the forward diffusion process. Denote the forward process as $q$ and the reverse process as $p$, the forward process starts from the original data $X_0$ and incrementally adds Gaussian noise to the data:

$$q(X_t|X_{t-1}) = \mathcal{N}(X_t; \sqrt{\alpha_t}X_{t-1}, (1-\alpha_t)I) \tag{7}$$

where $\{\alpha_t\}_{t=0}^T$ are noise schedule hyperparameters, $T$ is the total number of timesteps. Due to the Markovian nature of the forward transition kernel $q(X_t|X_{t-1})$, we can directly sample $X_t \sim q(X_t|X_0)$ without reliance on any other $t$:

$$q(X_t|X_0) = \mathcal{N}(X_t; \sqrt{\overline{\alpha}_t}X_0, (1-\overline{\alpha}_t)I), \quad \overline{\alpha}_t = \prod_{i=1}^t \alpha_i \tag{8}$$

$$\Rightarrow X_t = \sqrt{\overline{\alpha}_t}X_0 + \sqrt{1-\overline{\alpha}_t}\epsilon, \quad \epsilon \sim \mathcal{N}(0, I) \tag{9}$$

The set of $\{\alpha_t\}_{t=0}^T$ and $T$ should guarantee that $\overline{\alpha}_T$ is almost equal to 0, ensuring that the data distribution $q(X_T)$ closely resembles the standard Gaussian distribution. The corresponding normal posterior can be computed as:

$$q(X_{t-1}|X_t, X_0) = \mathcal{N}(X_{t-1}; \mu_q(X_t, X_0), \sigma_q^2(t)I)$$
$$\mu_q(X_t, X_0) = \frac{\sqrt{\alpha_t}(1-\overline{\alpha}_{t-1})X_t + \sqrt{\overline{\alpha}_{t-1}}(1-\alpha_t)X_0}{1-\overline{\alpha}_t} \tag{10}$$
$$\sigma_q^2(t) = \frac{(1-\alpha_t)(1-\overline{\alpha}_{t-1})}{1-\overline{\alpha}_t}$$

In training, the denoiser learns to predict the added noise $\epsilon$ in Equation 9 given noisy data $X_t$. The most commonly used objective function is to minimize the deviation between predicted noise and true noise:

$$L(\theta) = E_{t \sim U\{1,T\}, \epsilon \sim N(0,I)}||\epsilon - \epsilon_\theta(X_t(X_0, \epsilon), t)||^2 \tag{11}$$

Theoretically, it is equivariant to optimizing the evidence lower bound of $\log p(X_0)$:

$$L(\theta) \equiv E_{t \sim U\{2,T\}} E_{q(X_t|X_0)} D_{KL}(q(X_{t-1}|X_t, X_0)||p_\theta(X_{t-1}|X_t)) \tag{12}$$

During the reverse process, after substituting Equation 9 into Equation 10 and replacing the true noise $\epsilon$ with the predicted noise $\epsilon_\theta(X_t, t)$, we have:

$$p_\theta(X_{t-1}|X_t, t) = \mathcal{N}(X_{t-1}; \mu_\theta(X_t, t), \sigma_p^2(t)I)$$
$$\mu_\theta(X_t, t) = \frac{1}{\alpha_t}X_t - \frac{1-\alpha_t}{\sqrt{1-\overline{\alpha}_t}\sqrt{\alpha_t}}\epsilon_\theta(X_t, t) \tag{13}$$
$$\sigma_p^2(t) = \frac{(1-\alpha_t)(1-\overline{\alpha}_{t-1})}{1-\overline{\alpha}_t}$$

which defines the reverse transition kernel $p_\theta(X_{t-1}|X_t, t)$.

For the hyperparameters of DDPM, we adopt the linear noising schedule proposed by (Ho et al., 2021): $\{T = 1000; \quad \alpha_0 = 1 - 1e^{-4}; \quad \alpha_T = 1 - 2e^{-2}\}$.

## A.2 IMPLEMENT DETAILS

### A.2.1 DATA PREPROCESSING

We adopt the same configuration as the previous method: we first use the Ramer–Douglas–Peucker algorithm(Douglas & Peucker, 1973) with a parameter of $\epsilon = 2$ to remove redundant points. Then we maintain the aspect ratio and normalize the height of each sample to 1.

### A.2.2 MODEL PARAMETERIZATION

For the layout generator model, we adopt a 2-layer LSTM with a hidden size of 128 as the layout generator. *We also attempted to use a transformer as a replacement and found that the results were nearly the same*. Therefore, we chose the simpler and faster LSTM model. For the denoiser model, recall our model consists of a 1D U-Net network as the denoiser, a character embedding dictionary, and a multi-scale calligraphy style encoder. We set the dimension of character embedding as 150 and the dimension of time embedding for diffusion models as 32.

Figure 9 displays the interaction details between 1D U-Net denoiser and style encoder. Each downsampling block, such as $\{f^i, en^i\}_{i=1,2,3}$ consists of three residual dilated convolutional layers and a downsampling layer. The residual dilated convolutional layers have a kernel size of 3 and dilation values of 1, 3, and 5, respectively. The downsampling layer has a stride of 2 and a kernel size of 4. The upsampling blocks of the U-Net are structurally symmetrical to the downsampling block and maintain the same hyperparameters, with the only difference being the substitution of the convolutional layers in the downsampling blocks with deconvolutional layers. Table 4 illustrates the channel numbers of input and output features for each module. Afterward, we employ a simple linear layer to ensure that the output has the same dimensions as the initial data, thus performing the function of noise prediction.

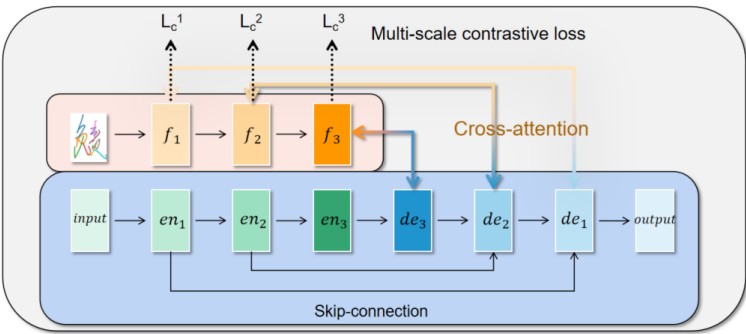

Figure 9: The framework of the multi-scale style encoder and the U-Net Denoiser. The number and stride of downsampling layers of U-Net and the style encoder are aligned, which we set as 3 in this work. The multi-scale style information is injected into U-Net through the cross-attention mechanism.

Table 4: Channel numbers of feature inputs and outputs at different layers..

|        | $f^1$ | $f^2$ | $f^3$ | $en^1$ | $en^2$ | $en^3$ | $de^1$ | $de^2$ | $de^3$ |
|--------|-------|-------|-------|--------|--------|--------|--------|--------|--------|
| input  | 128   | 256   | 512   | 128    | 256    | 512    | 256    | 512    | 1024   |
| output | 256   | 512   | 1024  | 256    | 512    | 1024   | 128    | 256    | 512    |

### A.2.3 TRAINING DETAILS

Actually, the layout planner module and the character synthesizer can be trained jointly. The size information that the layout planner predicts for each character is used as input to the single-character synthesizer, which is expected to generate characters with specific sizes. However, we find that not normalizing the character sizes will affect the model to learn structural information about the characters, leading to unstable generation results. Our approach is to decouple the training process of the layout planner module and the character synthesizer. When training the character synthesizer, we still first normalize all single characters to a fixed height. In this way, the synthesizer only needs to learn to generate characters at a standard size. We take full advantage of the nature that online handwriting data has no background noise, allowing us to directly scale the generated standard-sized characters and fill them into their corresponding bounding boxes.

We implement our model in Pytorch and run experiments on NVIDIA TITAN RTX 24G GPUs. Both training and testing are completed on a single GPU. For training the layout planner, the optimizer

is Adam with an initial learning rate of 0.01 and the batch size is 32. For training the diffusion character synthesizer, the initial learning rate is 0.001, the gradient clipping is 1.0, learning rate decay for each batch is 0.9998. We train the whole model with 400K iterations with the batch size of 64, which takes about 4 days.

### A.2.4 IMPLEMENTATION OF EVALUATION METRICS

**Style Score and Content Score:** For individual character evaluation, due to the limited writing styles represented by individual characters, following previous work(Dai et al., 2023; Tang & Lian, 2021), we combine fifteen characters written by the same author together as input. We train a style classifier with the task of distinguishing 60 writers in the test set, achieving an accuracy of 99.5%. The Style Score refers to the accuracy of classifying generated samples that imitate different styles using this classifier. The higher the style score, the better the model's ability to imitate calligraphy style. Similarly, we utilize all individual character data to train the Chinese character classifier, achieving an accuracy of 98%. Content Score refers to the accuracy of using this classifier to classify the generated samples. The higher the content score, the better the model's ability to generate accurate character structures. Both the style classifier and the content classifier adopt a 1D Convolutional Network, which is the same architecture as the style encoder introduced before.

**Accurate Rate and Correctness Rate:** Unlike single characters, when recognizing text lines, the number of characters within them is unknown. As a result, there may be discrepancies between the total number of characters and the number of correctly recognized characters in the content parsed by the recognizer. Therefore, in the absence of alignment, it is not appropriate to simply use the ratio of correctly recognized characters to the total number of characters as a measure of content score. Instead, evaluation metrics based on edit distance are commonly used:

$$CR = \frac{N_t - D_e - S_e}{N_t} \times 100\% \tag{14}$$

$$AR = \frac{N_t - D_e - S_e - I_e}{N_t} \times 100\% \tag{15}$$

where $N_t$ represents the total number of characters in the real handwritten text line, while $S_e$, $D_e$, and $I_e$ respectively denote substitution errors, deletion errors, and insertion errors.

### A.2.5 MORE VISUALIZATION RESULTS

For character generation, we have added a visual comparison between our method and the state-of-the-art style transfer method SDT Dai et al. (2023). As shown in Figure 10 our method achieves comparable performance and effectively mimics the target calligraphy style.

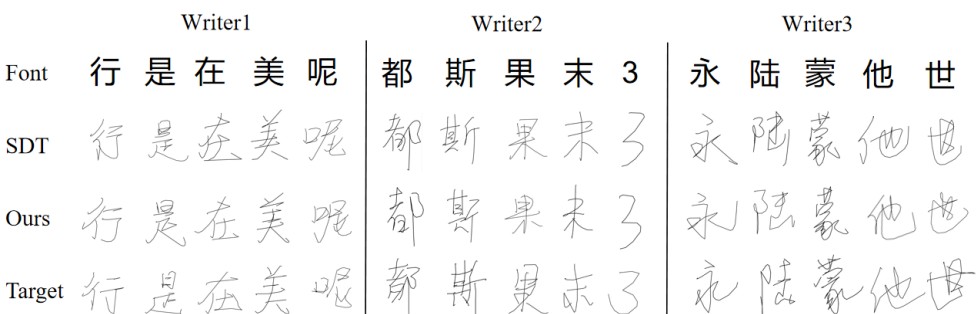

Figure 10: The visual comparisons with the state-of-the-art method SDT.

In Figure 11, For full text line generation, we select three authors with distinctly different writing styles as imitation targets. We visualized the real samples alongside the generated counterparts. The left columns display the real samples and the right columns display the generated data.

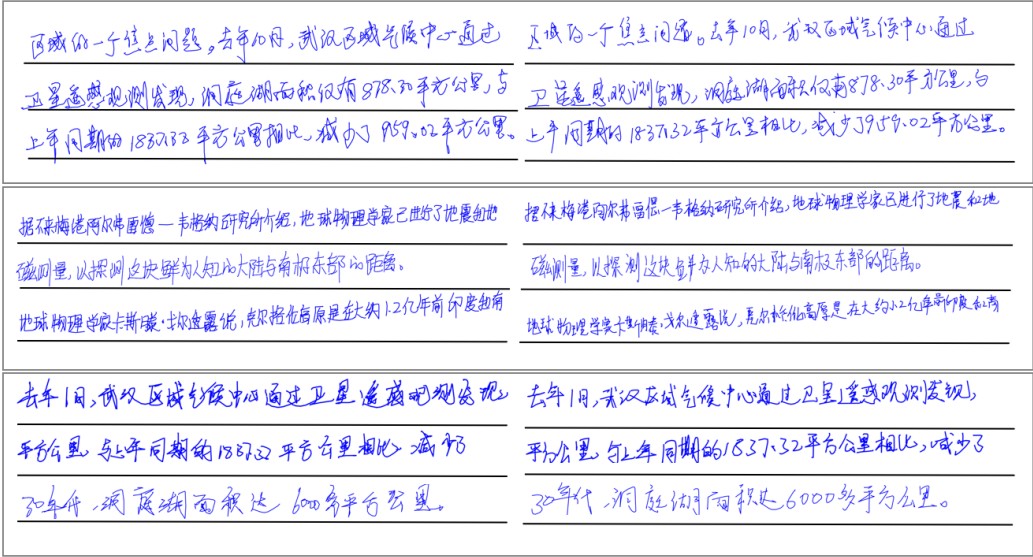

Figure 11: The visualization results. In each block, the left columns display the real samples which are the imitated targets of generated data on the right.

### A.3 THE POSSIBILITY OF EXTENDING TO AN END-TO-END APPROACH

We demonstrate the results of **directly using our proposed 1D U-Net** generation module by concatenating individual character embeddings into a full-line embedding for end-to-end generation. As shown in Figure 12, we specifically selected a handwriting style with connected strokes between characters, and the end-to-end model is capable of handling this. This also illustrates the potential of **extending our proposed method to an end-to-end generation framework**, demonstrating the generalization of our module.

Figure 13 shows the failure case, specifically, as the number of characters increases, structural errors tend to occur. In contrast, the decoupled layout and glyph method offers more stable training and inference. The next challenge will be to enhance the stability and ensure a more reliable generative process for end-to-end generation, which will be the focus of our future work.

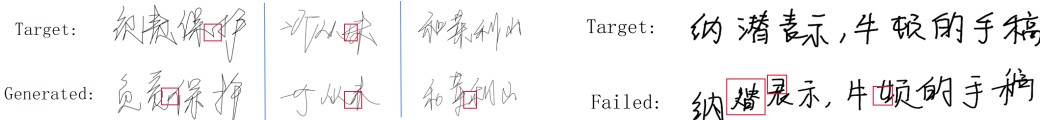

Figure 12: The end-to-end generation paradigm can produce connective strokes between characters.

Figure 13: The instability of directly using the end-to-end method with longer character sequences, resulting in failure cases.

## A.4 EFFECTS OF LAYOUT GENERATOR

We also tested using a portion of another text line written by the same author as a prefix to condition the layout generator for generation, with the results shown in Figure 14. It can be observed that the layouts generated by our proposed method still closely approximate the ground truth. This is because different samples from the same author typically share a consistent writing style. The key principle of our method is to ensure that the generated layouts align with the style of the input prefix, rather than requiring the prefix to be from the ground truth text lines. In contrast to previous method, our contributions and advantages lie in:

1. Compared to previous model-free method, our approach is capable of generating layouts that are closer to the real target. The reason is that model-free methods **deeply rely on manually designed rules and features**, which, especially in the case of Chinese text lines, are challenged by the varying sizes and shapes of different types of Chinese characters, as well as the significant differences in punctuation marks. This makes it difficult for earlier method to simultaneously capture the layout style. In contrast, our method, based on the next-token prediction paradigm, allows the model to naturally generate a layout where not only the positions of characters are reasonable, but the overall style of the layout is consistent with the given prefix, in a in-context manner.

2. In addition, in terms of generalization, the layout generator can consider both the horizontal and vertical relative positional relationships between characters, therefore is capable of handling simple 2D mathematical expressions, such as $e^x$ and $\frac{1}{2}$ without any modifications, while model-free methods are unable to achieve.

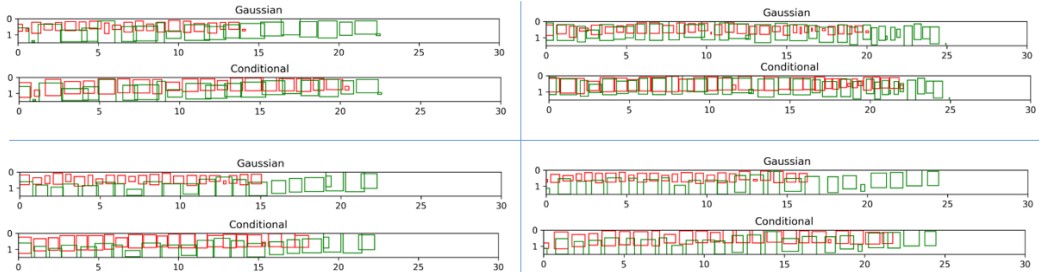

Figure 14: Layout generation visualization, where the prefix is taken from other text lines by the same author. The previous method is denoted as 'Gaussian'.

We have supplemented the quantitative experiments on layout generation presented in Table 5 as follows, where In-Context-gt refers to the case where we use the first 10 bounding boxes from the ground truth text line as the prefix, while In-Context-other refers to the case where we use 10 bounding boxes from another randomly picked text line written by the same author as the prefix. As can be seen, the performance is almost identical. The result suggests that as long as the layout style across different lines from the same author remains consistent, our method can perform effectively.

|  | $\nabla_1$ | $\nabla_2$ | $\nabla_3$ | $\nabla_4$ | $\nabla_5$ | $\nabla_6$ | $\nabla_7$ | $\nabla_8$ |
|---|---|---|---|---|---|---|---|---|
| In-Context-gt | 0.046 | 0.122 | 0.062 | 0.058 | 0.129 | 0.129 | 0.364 | 0.419 |
| In-Context-other | 0.047 | 0.124 | 0.057 | 0.058 | 0.130 | 0.132 | 0.363 | 0.422 |

Table 5: The performance quantitative evaluation results of the layout generator using other lines from the same author as prefixes.

