# Supplementary Material

## 1 Supplementary comparative experiment

Based on the suggestion, we have added a visual comparison between our method and the state-of-the-art style transfer method SDT. As shown in Figure 1 our method achieves comparable performance and effectively mimics the target calligraphy style.

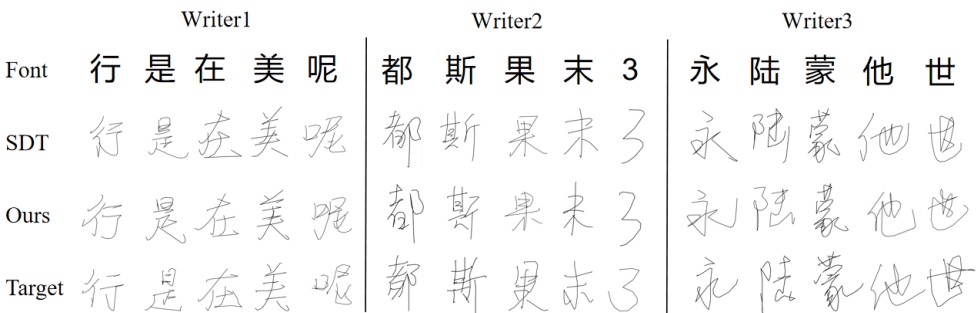

Figure 1: The visual comparisons with the state-of-the-art method SDT.

## 2 Revised Figure and visual experiment

**Revised Figure.** As shown in Figure 2, we have revised Figure 7 in the original manuscript, making it easier to identify which one corresponds to our method.

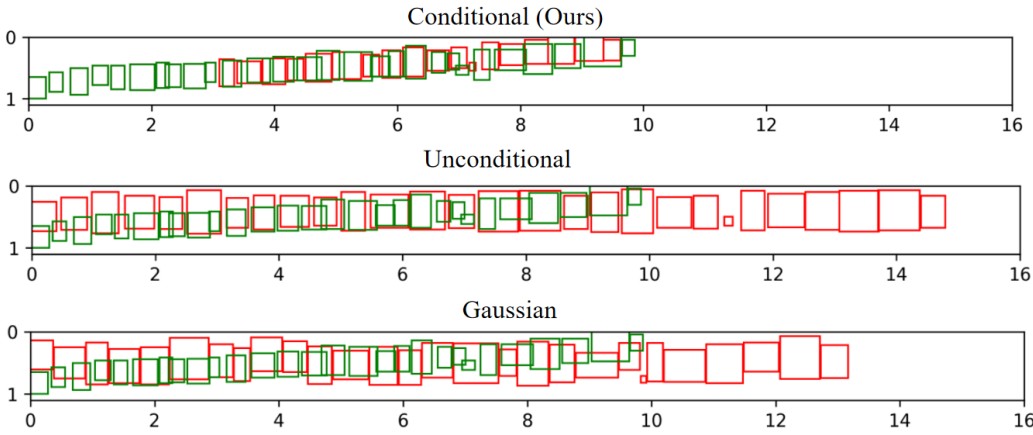

Figure 2: Revised layout case-study figure.

**Revised Experiment.** In Figure 3, we have included additional visual results, while also showcasing examples from the revised subjective experiment. Here, we present 8 different writers, with one line from a real sample and two lines from synthesized samples in each block. It can be observed that there is consistent style both within the synthesized samples and between the synthesized and target samples. Fifteen pairs of similar samples are created, and 20 participants are asked to determine whether lines in one block were written by the same person. Over 85% of the responses are "yes," indicating that the distinction between real and synthesized samples is difficult to perceive.

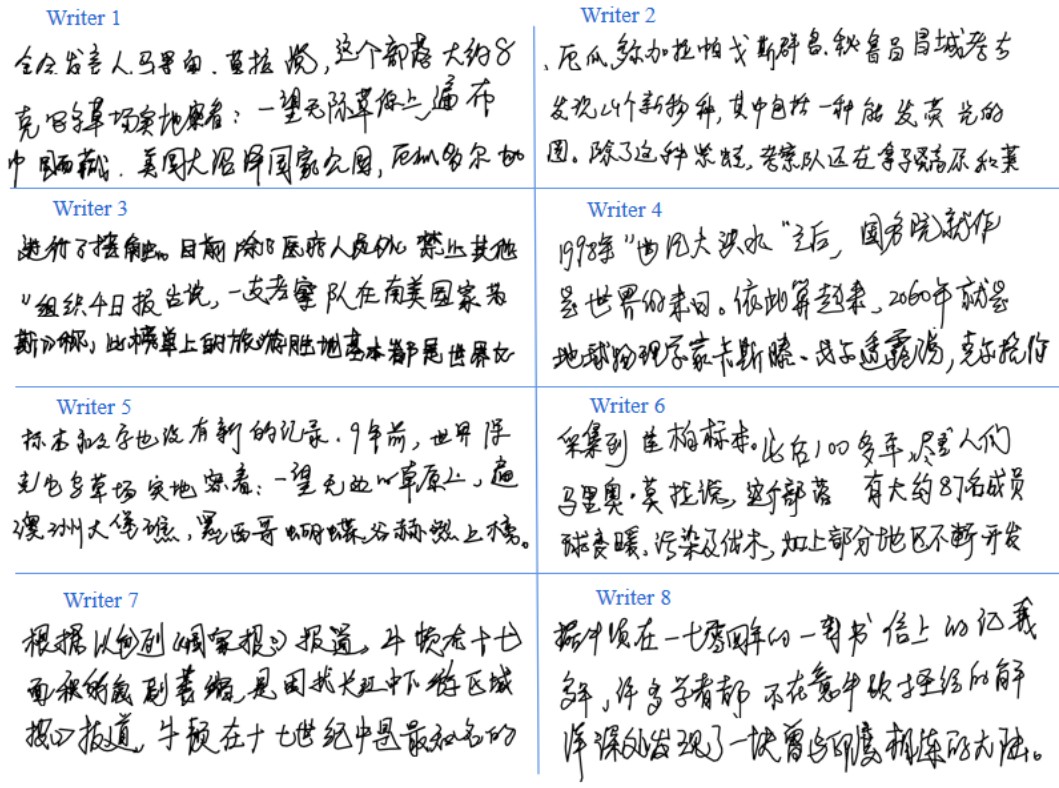

Figure 3: Revised subjective test figure.

## 3 THE POSSIBILITY OF EXTENDING TO AN END-TO-END APPROACH

We demonstrate the results of **directly using our proposed 1D U-Net** generation module by concatenating individual character embeddings into a full-line embedding for end-to-end generation. As shown in Figure 4, we specifically selected a handwriting style with connected strokes between characters, and the end-to-end model is capable of handling this. This also illustrates the potential of **extending our proposed method to an end-to-end generation framework**. Figure 5 shows the failure case, specifically, as the number of characters increases, structural errors tend to occur. In contrast, the decoupled layout and glyph method offers more stable training and inference. The next challenge will be to enhance the stability and ensure a more reliable generative process for end-to-end generation, which will be the focus of our future work.

Target: (handwritten Chinese) | Generated: (handwritten Chinese)

Target: 纳 潸 吉示，牛 报 的 手稿
Failed: 纳 熠 展示，牛 顿 的 手稿

Figure 4: The end-to-end generation paradigm can produce connective strokes between characters.

Figure 5: The instability of directly using the end-to-end method with longer character sequences, resulting in failure cases.

## 4 Effect of Layout Generator

As shown in Figure 6, we provide additional visual comparisons of the generated layouts with the existing method (denoted as Gaussian). For our method, the bounding boxes of the first ten characters are treated as the context prefix. In contrast to previous method, our contributions and advantages lie in:

1. Compared to previous model-free method, our approach is capable of generating layouts that are closer to the real target. The reason is that model-free methods **deeply rely on manually designed rules and features**, which, especially in the case of Chinese text lines, are challenged by the varying sizes and shapes of different types of Chinese characters, as well as the significant differences in punctuation marks. This makes it difficult for earlier method to simultaneously capture the layout style. In contrast, our method, based on the next-token prediction paradigm, allows the model to naturally generate a layout where not only the positions of characters are reasonable, but the overall style of the layout is consistent with the given prefix, in a in-context manner.

2. In addition, in terms of generalization, the layout generator can consider both the horizontal and vertical relative positional relationships between characters, therefore is capable of handling simple 2D mathematical expressions, such as $e^x$ and $\frac{1}{2}$ without any modifications, while model-free methods are unable to achieve.

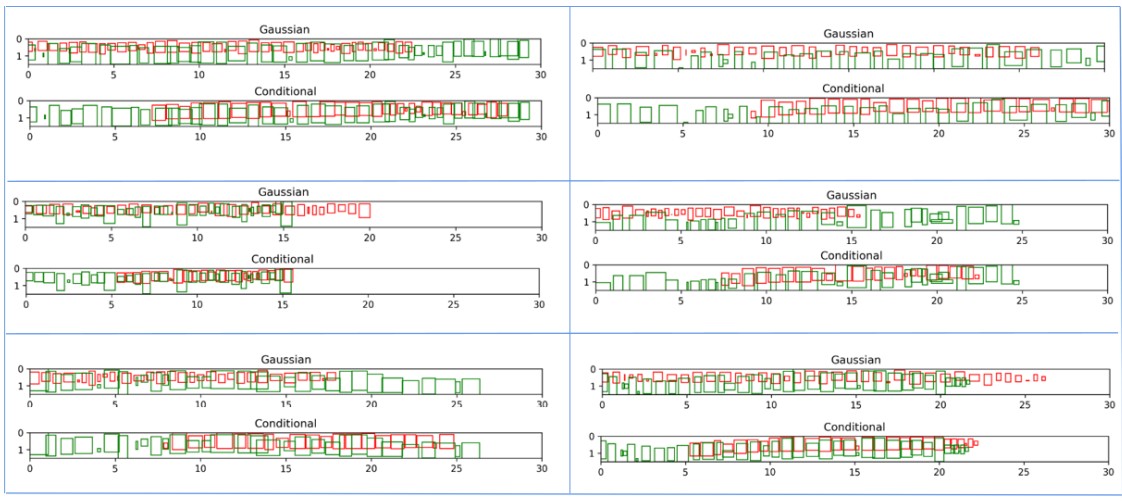

Figure 6: More visualization results of generated layout, where green represents the ground truth.

In addition, we also tested using a portion of another text line written by the same author as a prefix to condition the layout generator for generation, with the results shown in Figure 7. It can be observed that the layouts generated by our proposed method still closely approximate the ground truth. This is because different samples from the same author typically share a consistent writing style. The key principle of our method is to ensure that the generated layouts align with the style of the input prefix, rather than requiring the prefix to be from the ground truth text lines.

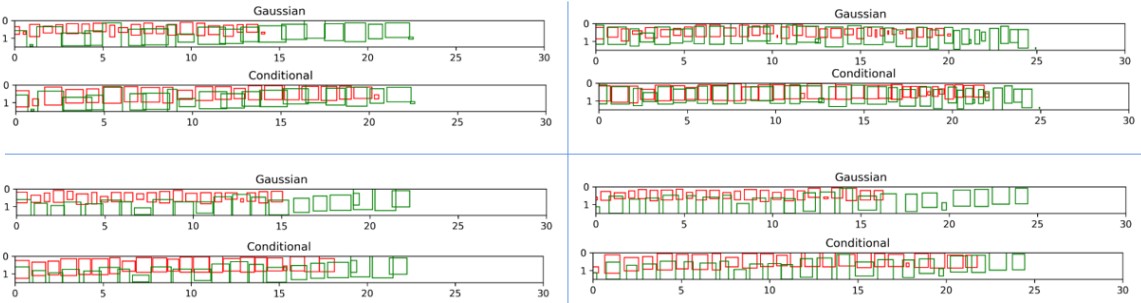

Figure 7: Layout generation visualization, where the prefix is taken from other text lines by the same author.