# OpenReview forum: "Decoupling Layout from Glyph in Online Chinese Handwriting Generation"
_ICLR.cc/2025/Conference — ICLR 2025 Poster_

### Official Review · Reviewer_Up7T · 2024-11-01

**Soundness:** 4
**Presentation:** 3
**Contribution:** 4
**Rating:** 8
**Confidence:** 5

**Summary:**

The paper addresses the task of generating online handwritten Chinese text lines condition on the content and style. It identifies that text lines can be divided into two components: layout and characters. The authors propose a hierarchical approach that includes a text line layout generator and a stylized font synthesizer. The layout generator uses in-context-like learning to determine the positions of each character, while the font synthesizer generates characters that imitate the calligraphic style of the provided references. The method is evaluated using the CASIA-OLHWDB dataset, demonstrating its effectiveness in producing structurally correct and indistinguishable imitation samples.

**Strengths:**

1.While some work on English handwritten text line generation exists, as far as I know, no such work has been published for online Chinese text lines. Compared to English characters, Chinese characters have more complex structures and a larger number of categories, making English generation methods unsuitable for direct application to Chinese. This work proposes a method to address this task, representing a noticable contribution.

2.The method decouples text line generation into two steps—layout generation and character generation—under a unified probabilistic framework, providing a good theoretical foundation and considerable novelty.

3.The experimental section includes comprehensive comparative and visualization experiments for both layout generation and character generation, yielding convincing results.

4.The paper is well-organized and clearly written.

**Weaknesses:**

1. The assumption that character generation is independent given their positions seems too strong. Does text line style only manifest in the relative positions and sizes of individual characters? I hope the authors can give discussion on the reasonableness of this assumption and explain whether it might limit the method's ability in style learning.

2. It is better to add sub-figure index for figure 8 and 9. It seems each of figure 8 and 9 has three sub-figures, but now their boundaries are not clear. In Figure 7, it is also suggested to identify which one is the proposed method in the paper. Of course, this is not a big issue.

**Questions:**

1. If the bounding box generated by the layout model and the bounding box generated by the character model have different shapes, how should this be handled?

2. Since the method can be described as a unified probability distribution according to Equation 1, why not jointly train the two models end-to-end instead of training them separately?

3. The paper does not discuss whether the method can be applied to handwriting generation of other languages.

4. in Line 285, what does L represent? Although the authors write this is the length of the feature sequence, it is not clear what does this sequence represent?

5. In Line 230, is the ground truth the same as the reference input in Figure 3？

---

> ### Author Response · Authors · 2024-11-14
>
> We truly appreciate the positive feedback and recognition of our work’s novelty, contribution and foundation！In the following, we provide detailed responses to the queries one by one:
>
> ---
>
> >Comment1：Does text line style only manifest in the relative positions and sizes of individual characters? The reasonableness of the indepent assumption and explain whether the decoupling might limit the method's ability in style learning.
>
> **Answer**:
> Thank you for raising such a valuable and thought-provoking question. This issue is indeed worth exploring further and may provides important insights for future improvements to our method.
>
> + On the manifestation of text line style: We agree that the entire text line writing style is difficult to be fully captured with disentangled styles. However, we believe that the calligraphy style of individual characters, as well as their relative positions and relationships, form the most intuitive and crucial components of the entire style. In fact, these factors are sufficient to describe **the vast majority** of handwriting styles, making them central to our approach.
>
> + On the reasonableness of the independence assumption: Our independence assumption is based on extensive observations of writing habits and visualizations from the dataset. We believe this assumption holds in most practical scenarios. This is ultimately an empirical question, and the validity of our assumption can be partially supported by the subjective satisfaction observed in the user studies of synthesized samples.
>
> + On the decoupling and potential limitations: We acknowledge that decoupling the writing process does introduce some limitations. The most powerful solution may indeed lie in end-to-end manner. However, training a high-quality, end-to-end Chinese text line generation model remains a challenging task. As such, our current approach represents a trade-off between simplicity and performance.
>
> >Comment2：About some figure issues.
>
> **Answer**：Thank you for your helpful suggestion and appreciate your attention to these details, and we will make these revisions in the revised version of the paper.
>
> >Question1&2：1. If the generated bounding box have different shapes with the generated characters, how should this be handled? 2. Why not jointly train the two models end-to-end instead of training them separately?
>
> **Answer**：The two issues are interrelated, so I will address them together:
> +  Referring to Appendix 2.1, in the data preprocessing stage, we normalize the overall height of text lines to 1, which results in **significant variance in the actual size of each character**. For instance, in lines with horizontal writing, character heights are close to 1, but in lines with certain tilted writing angles, character heights may be as low as 0.3. As stated in Appendix 2.3 (training details): ”In practical implementation, we found that without normalizing the size of the characters, the model's ability to learn the structural information of the characters would be compromised, leading to consistent errors in the generated structures. Therefore, during the training of the character generator, each character is normalized to the same size and learned independently.“
> +  As described above, the character generator directly produces characters with normalized size. We employed a simple method of scaling the xy-coordinates to fit the bounding box output by the layout generator. Since the layout generator takes into account the types of characters, the bounding boxes it generates are reasonable. Consequently, we observed that this scaling does not significantly affect the shape or aspect ratio of the characters.
>
> >Question3: Whether the method can be applied to other languages?
>
> **Answer**：Due to the decoupling of layout and glyph properties, our method is especially well-suited for language systems in which characters are relatively independent, such as Chinese, Japanese, Korean, or even mathematical formulas.
>
> >Question4:   What does 'L', 'sequence' represent in line 285?
>
> **Answer**：Recalling section3.1, our data consists of online **sequential data**. For example, if there are 1000 trajectory points, the raw data dimension would be $\mathbb{R}^{(1000,3)}$. Our network consist of 1D convolutional layers, for example, after passing through a convolutional layer with a stride of 2 and a kernel num of  D, the feature dimension would be reduced to $\mathbb{R}^{(500,D)}$.  L represents the length of the feature sequence, which is 500 in this case.
>
> >Question5: In Line 230, is the ground truth the same as the reference input in Figure 3？
>
> **Answer**：Yes, we use the **teacher-forcing** technique during training, where for each text line, the bounding boxes of the first i-1 characters are used as the prefix (reference) to predict the bounding box of the i-th character.
>
> ---
> Thank you  again for your insightful suggestions to improve our paper! We hope that our response adequately addresses your concerns.

---

> > ### Author Response · Authors · 2024-11-17
> >
> > Dear Reviewer,
> >
> > We have just submitted the supplementary material. In response to your questions and suggestions, we have made the following revisions:
> > +  In Section2, we have revised Figure 7 in the original paper, making it easier to identify which one corresponds to our method.
> > For Figures 8 and 9 in the original paper, we have also revised the visualizations to make the boundaries between different authors more distinct. Thank you again for your valuable suggestion!
> > + In Section3, we present the progress of extension experiments using our designed network, aimed at addressing the end-to-end generation paradigm for handling cursive character connections.
> >
> > Thank you for your recognition of our work! We warmly welcome further discussion if you have any additional questions or suggestions!

---

> > > ### Comment · Reviewer_Up7T · 2024-11-18
> > > **follow-up questions**
> > >
> > > Thanks for the detailed responses, which address my most of concerns. However, I am still not clear about the following three questions.
> > >
> > > (1) Do authors consider the direction of each generated character? For example, I think the italic character in each position is very common.
> > >
> > > (2) In line 285, it seems the L represents the length of w_i (the i-th character, please correct me if my understanding is wrong). So each character has 500 trajectory points? why each character have the same number of points?
> > >
> > > (3) I am sorry that maybe I did not describe my question clearly. In the paper, the authors describe that "use l1 distance between ground truth and generated layout as the loss function". so my question is that how to get the ground truth here? Is this the same as the style reference during the training? in Figure 3? Or the dataset has exact ground-truth annotation for each generated layout?

---

> ### Author Response · Authors · 2024-11-18
> **answers**
>
> Thank you for your detailed question！ Below are the answers to the individual queries:
>
>
> (1) In the layout generation stage, we donot explicitly consider the slant (italic) of the generated characters, as we believe this factor can be **addressed in the second stage** of individual character generation. If all provided style reference samples exhibit the same slant, the generated character samples will similarly imitate this slant. For example, in Figure 3 of the supplementary material, "writer2," both the real and generated characters exhibit a similar slant angle.
>
> (2) Yes, your understanding is correct. To be more precise, the L in line 285 represents the length corresponding to the i-th character at a certain feature layer. (The value 500 is just an example to illustrate the meaning of L, and not every character has this many points.)
>
> (3) Thank you for the clarification. The dataset is annotated. More specifically, the dataset labels which trajectory points belong to each single character. Then, following the definition of layout described in Section 3.2 of the original paper, we manually calculate these ground truth values during the data processing phase.
>
> Based on the previous discussion, we have made revisions and additions to the original text. And we hope our response effectively addresses your concerns！

---

> > ### Comment · Reviewer_Up7T · 2024-11-22
> > **Following question (3)**
> >
> > Thanks for your clarification. For the Q(3), based on my understanding, the output text line will match both style reference and text content. If the dataset has ground-truth of each generated text line, does it mean the ground-truth also match the reference style？ So the data has such kind of triplet annotation <text content, style reference, ground-truth of generated text line>?

---

> ### Author Response · Authors · 2024-11-22
> **answers**
>
> Thank you for your detailed question! Here, we will provide a detailed introduction to the dataset and the specifics of its usage：The dataset can first be divided into 1,200 different authors, with each author having dozens of handwritten lines.
> ***
> Therefore, for each author i, we have all the text lines that he has written:
>
> ${writer_i}: [(line_{i1}, content_{i1}), (line_{i2}, content_{i2}),..., (line_{in}, content_{in})]$
>
> Assume that we use the j-th line from author i as the imitation target, so $line_{ij}$ serves as the ground truth. For the style reference, we randomly select  the **k-th line (where k ≠ j) from the same author i** as the style reference $ref_{style-i}$. Thus, the generative model's input is composed of <$content_{ij}$, $ref_{style-i}$ > and its target ground truth is $line_{ij}$.
>
> ***
> In summary, for each text line used as the imitation target, other text lines from the same author can be used as style references. Therefore, the ground truth matches the style reference sample, as they originate from the same author. Hope this explanation can resolve your issue!

---

> > ### Comment · Reviewer_Up7T · 2024-11-22
> >
> > Thanks for the clarification. This is a good idea. It is clear now and I have no further questions.

---

> > > ### Author Response · Authors · 2024-11-22
> > >
> > > We sincerely appreciate your response and the recognition of our work! This is of great significance to the continued improvement of our efforts.
> > >
> > > Best wishes！

---

### Official Review · Reviewer_BU7e · 2024-11-02

**Soundness:** 3
**Presentation:** 2
**Contribution:** 2
**Rating:** 5
**Confidence:** 5

**Summary:**

This paper focuses on the generation of online Chinese handwritten text lines. The core of this method lies in decomposing text line generation into layout generation and character generation, and fill characters into the generated layouts to form complete text lines. Experiments evaluate the proposed method.

**Strengths:**

1) This paper proposes a hierarchical online Chinese handwritten text line generation method. The proposed method utilizes a layout generator and a font synthesizer to produce the layouts and characters independently, then arranges the characters within the layouts to create complete text lines.

2) The proposed method achieves the best performance in purely data-driven font generation task.

**Weaknesses:**

1) The multi-scale style encoder is not a new design in handwriting generation area, as a similar idea has been proposed in [a]. Besides, the proposed style contrastive learning loss is somewhat similar to the style learning loss in [b].

2) The method description is not clear: (1) In lines 233-237, it is mentioned that style reference samples are used as context prefixes, but how they guide the subsequent layout generation is unclear. (2) The paper does not specify the modality of the style references used, online data, or offline images. (3) The paper does not specify the number of style reference samples used, one-shot or few-shot.

3) Section 4.3.2 lacks quantitative experiments in terms of calligraphy styles, raising doubts about whether the proposed Multi-Scale Style Encoder can accurately extract calligraphy styles from entire text lines.

4) In the 'Conditional' row of Figure 7, the generated layouts (red boxes) show significant absences at the beginning of the text line, which raises concerns about the effectiveness of the layout generator.

5) It is recommended to compare the proposed method with style transfer-based approaches, as it can be relatively straightforward to extend this method to a style transfer setting by replacing character embedding with a CNN-based content encoder.

6) The layout generator requires real layouts of style references, which is not directly available in the application, does this limitation affect its applicability? If some simple layout extraction methods are used to extract the pseudo-layouts of style references, what impact would this have on generation performance?

7) The paper provides very few generated visual results and lacks visual comparisons with the baseline.

[a] Wang H, Wang Y, Wei H. Affganwriting: a handwriting image generation method based on multi-feature fusion, ICDAR, 2023.

[b] Dai G, Zhang Y, Wang Q, et al. Disentangling writer and character styles for handwriting generation, CVPR, 2023.

**Questions:**

My main concerns are the novelty of the proposed multi-scale style encoder and style contrastive learning loss, and the effectiveness of the proposed layout generator. For details, please refer to the weaknesses.

---

> ### Author Response · Authors · 2024-11-13
> **Response to Concerns Regarding the Novelty and the Effectiveness of the Proposed Method.**
>
> Thank you very much for your comments and questions, which are of great significance for us to improve our article and work. Next, I will respond in detail.
>
> ### **Regarding the novelty and contribution** :
> Firstly, we would like to emphasize that the primary novelty of our work lies in the proposal of a hierarchical approach to **solving the challenging task of handwritten Chinese text line generation**, a problem that has been **rarely explored**. This decoupling strategy can also be effectively extended to other complex handwriting generation tasks. Our contributions within this framework are twofold: 1) A novel method for explicit layout modeling, and 2) A purely data-driven character generation approach based on a 1D U-Net.
>
>
> For 1)： I would like to highlight that we cleverly adapt the in-context generation paradigm from **next-token prediction** in LLM to the task of generating layouts. This simple yet effective approach forms the core of our method. To the best of our knowledge, our work is one of the earliest explorations into handwritten Chinese text line generation, and it is the **first to explicitly generate the layout for text line**. Additionally, an extra benefit of our approach is that the generated data **naturally contains strong positional label** of characters, which makes it highly convenient for data augmentation in tasks such as character segmentation and recognition.
>
> For 2）: We design a 1D convolutional network capable of extracting **multi-scale features** specifically for **online sequential data**. In contrast, previous approaches for sequential handwriting data typically use LSTM or Transformer models, which do not explicitly consider multi-scale features. Furthermore, our approach is purely data-driven, distinguishing it from the style transfer paradigm commonly used in the state-of-the-art methods. These two approaches are not mutually exclusive but can complement each other in different application scenarios.
>
> Below, we address the reviewer’s questions one by one:
>
> ---
>
> ### **Response to Specific Comments** :
> > Comment 1: About the novelty of the multi-scale encoder and multi-scale contrastive loss.
>
> **Answer** : Thank you for pointing out the relevant literature. The additional references have been incorporated in our revision. However, it is important to note that the work referenced in [a] deals with offline handwritten data, specifically images. In contrast, all the data used in our work is online data, which is sequential in nature. Networks designed for processing **online handwritten data** typically **do not explicitly** model multi-scale features in the previous related work. In this paper, we introduce a model based on 1D-convolutional networks, marking the first instance of explicitly modeling multi-scale features in online Chinese handwritten data generation field. Additionally, the contrastive loss function we design is tightly integrated with this multi-scale network to enhance its ability to distinguish calligraphy styles from different writers at different scales. While in paper[b], the proposed writer-wise and character-wise contrastive learning is applied **only at a single feature scale**. From this perspective, our contributions are not similar, but they could even **complement each other**.
>
> > Comment 2:  It is not clear : 1) How the prefix guide the generation in layout generator.  2) The modality of style reference data.  3) The task setting.
>
> **Answer** :  1) A more intuitive explanation can be drawn from the **next-token-prediction paradigm** in language models. When a partial sequence of text is input as a context, the subsequent generated content tends to be coherent with the prefix. Similarly, in our task, when the layout information for the first few characters is input as a context, the layout of subsequent characters is naturally consistent with the prefix. For example, if the initial layout exhibits a general skew, there is a higher probability that this characteristic will be maintained in the generated text, as we demonstrate in the experiment section (Figure 7).  This is what we call **“in-context-like layout generation”** .  2) As described in Section 3.1, all the data we used in this paper is online sequential data. Therefore, the novelty of our approach lies, to some extent, in its focus on designing a network specifically tailored for online handwritten data compared with previous work. 3) For fairness in comparison, we have kept the experimental setup consistent with previous methods, specifically using a few-shot setting, where 10 reference characters are used for style reference.
>
>
> [a] Wang H, Wang Y, Wei H. Affganwriting: a handwriting image generation method based on multi-feature fusion, ICDAR, 2023.
>
> [b] Dai G, Zhang Y, et al. Disentangling writer and character styles for handwriting generation, CVPR, 2023.

---

> ### Author Response · Authors · 2024-11-13
>
> > Comment 3:  Section 4.3.2 lacks quantitative experiments.
>
> **Answer**: Actually, our calligraphy classifier **can also achieves a classification accuracy of 91%** for the entire generated text line. The reason for this is not explicitly included in the paper is that, in Chinese, calligraphy style can be predominantly reflected at the level of individual characters. Since our method generates each character independently, the quantitative evaluation of calligraphy style of the entire line **largely overlaps with** the character-level experiments presented in Section 4.2.
>
> > Comment 4:  The generated layouts show significant absences in Figure 7.
>
> **Answer**: Recalling that our layout generation method requires a few characters bounding boxes as a prefix. In Figure 7, we use the first ten characters as the prefix, so the bounding boxes for these characters are the same as the real ones rather than being missing.
>
> > Comment 5:  It is recommended to compare the embedding-based method cnn-content-encoder approaches.
>
> **Answer**:  Thanks for your insightful suggestion. We conducted a supplementary experiment for Table 1 by replacing the character embedding with the cnn-based content encoder：
>
> |               | **DTW(↓)** | **Content Score(↑)** | **Style Score(↑)** |
> | ------------- | :-----: | :---------------: | :-------------: |
> | **CNN-based** |  0.943  |      0.935        |      0.892      |
> | **Ours**      |  0.932  |      0.891        |      0.918      |
>
>
> It can be observed that the content score has increased, while the style score has slightly decreased, with a small increase in computational cost. Overall, the performance of both methods is comparable. We believe that the advantage of style transfer-based methods lies in their ability to generalize to standard glyphs that were not seen during training, whereas embedding-based methods can handle datasets that lack standard glyphs.
> + As mentioned before, these two settings are not conflict ing but can complement each other in different application scenarios.
> + Additionally, in Section 3 of the supplementary materials, we demonstrate the flexibility of our approach in end-to-end text line generation, where **the content encoding of a text line can be naturally obtained by concatenating the embeddings of individual characters**.
>
> > Comment 6: 1) About the applicability and 2) what will happen if some simple layout extraction methods are used to extract the pseudo-layouts of style references.
>
> **Answer**:   1) Application:
> + For customized handwritten text generation, we only require the user to write a small piece of coherent text line (e.g 10 characters) and mark the position of each char. It do not need to be coherent in content with the text lines to be generated later, but only used as a style reference. The layout generator will mimic the layout characteristics of these reference characters when generating the layout for subsequent characters. Additionally, these 10 reference characters will also serve as style references for the character generation model.
> + Last but not least, the strong positional label of characters in our generated data also makes it highly convenient for data augmentation in tasks such as character segmentation and recognition.
>
> 2）Performance of simple layout style extraction methods tends to be **overly dependent on** carefully designed features, such as the binary geometric features I used in Table 2. In my own experiments, I found that model-free approaches that are too simplistic often generate layouts with noticeable differences compared to real samples, such as failing to properly handle the relative positions of punctuation marks and text. Furthermore, these methods exhibit poor generalization and are **only limited to text line generation**, whereas model-based approaches can be transferred to other even 2-Dimension handwritten data, such as handwritten math equations.
>
> > Comment 7:  Few generated visual results and lacks visual comparisons with the baseline.
>
> **Answer**: Thank you for your suggestion. We **have conducted** more visulization and subjective experiments as well as visual comparisons with previous sota methods in the revised version to make our paper more convincing. However, in the domain of text line generation, this task remains largely unexplored, and as such, there is a lack of established baselines. Our approach represents a **novel attempt at addressing this task**.
>
> ---
> We sincerely appreciate your detailed and thoughtful feedback on our manuscript. We have carefully addressed each of your comments and have made the necessary revisions and additions in the official version of the manuscript.  We hope that our responces meet your expectations and would be grateful if you could consider revising the rating in light of the improvements made. We welcome any further questions or discussions you may have, and we will be happy to provide more elaborate responses as needed！

---

> ### Author Response · Authors · 2024-11-17
>
> Dear Reviewer,
>
> We have just submitted the supplementary material. In response to your questions and suggestions, we have made the following revisions:
> + In Section 1, we present a visual comparison with state-of-the-art methods, which makes our experimental results more comprehensive.
> + In Section 2, we have added additional visual results and further refined the subjective evaluation experiments.
> + In Section 3, we demonstrate the generalization of our proposed model to an end-to-end framework. We believe these updates further **highlight the novelty and flexibility** of our approach compared to existing work. This also demonstrates one advantage of embedding-based methods over CNN-encoder-based methods in terms of scalability, as style transfer-based approaches face challenges in obtaining standard content templates for **complete text lines**.
> + In Section 4, we present more visual comparison results of the layout generation methods with previous approaches and carefully elaborate on our advantages and contributions.
>
> We have also made these revisions as well as added the corresponding references to the relevant sections of the original paper. Thank you once again for your thoughtful feedback! If you have any unresolved issues or suggestions, please do not hesitate to let us know!

---

> ### Comment · Reviewer_BU7e · 2024-11-23
> **Response to authors**
>
> Thank you for your detailed responses, which partially address my concerns. However,
> 1) I still believe the multi-scale style encoder with contrastive loss is not a novel design within the handwriting generation domain. For the multi-scale style encoder, it incorporates the same multi-scale idea as [a], with the key difference being in the implementation: the proposed encoder uses a 1D CNN, while [a] employs a 2D CNN. This distinction, however, is a minor modification. Regarding the multi-scale contrastive loss, it essentially introduces the multi-scale concept of [a] into [b], i.e., applying contrastive learning loss across multi-scale features. Considering that both [a] and [b] are established works in handwriting generation, combining them does not constitute a novel design.
>
> 2) The performance of the updated CNN-based version appears to lag behind the previous style transfer method, SDT (cf. Table 1). As a result, the contribution of the proposed method to font generation (cf. Section 4.2) is limited to the purely data-driven setting.
>
> 3) The proposed method seems user-unfriendly, as it not only requires 10 reference characters but also forces users to manually mark the position of each character.
>
> Besides, my new questions are:
> 1) In Table 3, the metric AR is consistently higher than CR. However, according to the definitions in [c, d], AR includes insertion errors as an additional factor compared to CR, which should make it lower than CR. Could you clarify these results?
>
> 2) It appears that the first 10 characters of each text line are used as style references to predict the subsequent layouts (cf. the caption of Figure 7). However, these 10 characters seem to be part of the ground truth (GT). Does this lead to information leakage from the GT? In other words, are parts of the GT used as style references? Could you clarify this?
>
> Looking forward to your reply.

---

> ### Author Response · Authors · 2024-11-23
> **Response to the following questions**
>
> Thank you for the clarification on the issues! Below are the responses to your concerns and questions:
> > Comment 1: The novelty about the multi-scale style encoder with contrastive loss.
>
> **Answer**:
> Thank you for your insightful feedback based on your extensive domain knowledge. We will provide a more detailed explanation of our contribution:
> + **Background**: As previously stated, our method operates entirely on online data. Since online trajectories can contain more information, especially writing order, research on processing online trajectories is certainly worth investigating. However, previous methods  based on online data extract style features in a relatively crude way, such as they do not leverage multi-scale features of online data. Therefore, how to extract rich features from online data is a topic worthy of further research.
> + **Contribution**: We fully acknowledge your valuable insights in the relevant field, while the additional references have been incorporated in our revision. However, although similar concepts may exist in the related work on offline data, considering the background of online data, our approach is the **first implementation** to extract multi-scale style features from **online data** and has also demonstrated its effectiveness, provides a foundation for future research on style feature extraction for online data. We believe this is of certain value. More importantly, this is only a small part of our framework and not the main contribution.
>
> >Comment2: About the contribution of the proposed method to font generation.
>
> **Answer**:
>
> + **Background**: As mentioned before, style transfer-based methods and purely data-driven methods have different potential applications scenarios, and therefore cannot be completely replaced by each other. Moreover, the performance of the earlier purely data-driven methods **significantly lagged behind** that of style transfer-based methods. Therefore, how to improve the performance of purely data-driven methods in this field reamains an important research question.
>
> + **Contribution**:
>
>   1).  For font: In our work, we have significantly bridged the performance gap between purely data-driven methods and sota style-transfer methods, therefore laid the foundation for future studies on purely data-driven approaches. More specifically, our method performs almost identically in terms of style scores with the sota style-transfer methods, while it slightly lags behind in content scores. We attribute this primarily to the fact that purely data-driven methods, lacking standard font structure information, need to learn everything from scratch. As a result, they are more susceptible to annotation errors and noisy, overly sloppy samples in the dataset, which can negatively affect the stability.
>
>
>   2).  For generalization: More importantly, the 1D CNN-based denoiser we designed **can be directly extended for end-to-end multi-character generation** (see Appendix A.3 or Supplementary Material Section 3), which represents a **significant advantage** over previous methods. In contrast, previous methods for single-character generation typically focus on transferring a standard character template style to a handwritten style. However, text lines not only involve character structure but also include the size and position of characters, meaning they lack a standard template, making these methods unsuitable for text line generation tasks.
>
> >Comment3: About the application scenary.
>
> **Answer**:
> + **Writing service** :  The first point to emphasize is that we are the **first to provide a service** capable of generating text line-level data of arbitrary length while maintaining the user's layout style in a single pass. Additionally, in practical applications, users only need to write a continuous sequence of characters (less than 10 is also acceptable), which are then processed by a character segmentation algorithm to get the bounding box information we need. The current segmentation algorithm's performance is sufficient to meet the requirements of our application. The users do not need to mark the position. We believe that it is still very convenient for users to use.
>
> + **Other application**: In addition, our approach is not limited to providing handwriting services only. The generated data naturally contains strong positional label of characters, which makes it highly convenient for data augmentation in tasks suach as confidence calibration where character position is needed to determine positive or negative samples. We believe that our approach has significant potential for application in this regard.
>
> > Question1:  About the AR and CR.
>
> **Answer**:   Thank you for your attention to detail!  Apologies for the mistake in Table 3 where we swapped AR and CR and this has now been corrected now.

---

> ### Author Response · Authors · 2024-11-23
> **Response to the following questions**
>
> > Question2:  About the effectiveness of the layout generator.
>
> **Answer**:   Layout generation is the core of our approach and one of the most significant innovations of our method.
>
> It is important to emphasize that: **the effectiveness of our in-context layout generation method lies in its ability to ensure that the layout style of the generated portion remains consistent with that of the prefix portion, rather than whether the prefix belongs to the ground truth text line.**  To demonstrate this point, we have conducted further quantitative and qualitative experiments:
>
> 1. Quantitatively :We have supplemented the quantitative experiments on layout generation presented in Table 3 as follows: ( In-Context-gt refers to the case where we use the first 10 words from the ground truth text line as the prefix, while In-Context-other refers to the case where we use 10 words from another randomly picked text line written by the same author as the prefix.)  As can be seen, the performance is almost identical, because the style of different text lines written by the same author is nearly consistent.
>
> |            | $\nabla_1$ | $\nabla_2$ | $\nabla_3$ | $\nabla_4$ | $\nabla_5$ | $\nabla_6$ |$\nabla_7$ | $\nabla_8$ |
> |----------|----------|----------|----------|----------|----------|----------|----------|----------|
> | In-Context-gt    | 0.046 | 0.122 | 0.062 | 0.058 | 0.129 | 0.129 | 0.364 | 0.419 |
> | In-Context-other | 0.047 | 0.124 | 0.057 | 0.058 | 0.130 | 0.132 | 0.363 | 0.422 |
>
> 2. Qualitatively: In Figure 7 of the origin paper, we indeed use the first 10 characters from the ground truth text line as the prefix. This is primarily to provide a more intuitive demonstration that our method can maintain **consistency between the layout style of the generated part and the prefix part**. We have supplemented the qualitative experiments in the Section 4, Figure 7 of the supplimentary material. We demonstrate that using ten characters from other text lines from the same author as a prefix, our method can still capture the layout style quite effectively. As long as the layout style of different lines from one same author remain consistent, our method can work well.
>
> Thank you for your thoughtful questions, which have help us to make our experimental section more comprehensive.
>
> ---
> We hope that our response can effectively address your concerns. If you have any unresolved issues or suggestions, we would be more than happy to provide further clarification and make improvements to the best of our ability, as your suggestions have played an important role in improving our work!

---

> > ### Comment · Reviewer_BU7e · 2024-11-26
> > **Response to authors**
> >
> > Thank you for the detailed answer. It has addressed part of my concerns very well, and I am happy to raise the score to 5.

---

> > > ### Author Response · Authors · 2024-11-26
> > >
> > > Thank you for recognizing our efforts and improvements. Your rigor and expertise have provided us with valuable suggestions to enhance our work. We appreciate your contribution to the paper!
> > >
> > > Warm regards,
> > >
> > > The Authors

---

### Official Review · Reviewer_kxE9 · 2024-11-03

**Soundness:** 3
**Presentation:** 2
**Contribution:** 3
**Rating:** 6
**Confidence:** 3

**Summary:**

This paper focuses on the generation of online Chinese handwriting text lines. It proposes a hierarchical approach that decouples layout generation from glyph generation. The text line layout generator arranges character positions based on text content and writing style references, while the font synthesizer generates characters with specific styles. The contributions include a novel layout generator, a 1D U-Net network for font generation, and a multi-scale style encoder. Experiments demonstrate the effectiveness of the method in generating structurally correct and stylistically similar samples.

**Strengths:**

(1) The hierarchical decomposition into layout and glyph generation is an innovative approach, particularly suited for complex scripts like Chinese. This framework successfully addresses challenges specific to the language, such as the diversity of character structures.

(2) The model is thoroughly tested on both character and line generation, with metrics tailored to layout and stylistic fidelity. The model's success across multiple metrics shows a well-rounded, effective design.

(3) Despite the technical depth, the paper provides a good level of explanation for each module, with helpful visualizations that demonstrate layout and glyph generation separately.

(4) The method has potential applications in handwriting synthesis, digital personalization, and document augmentation, contributing a valuable approach for future research in multilingual handwriting generation.

**Weaknesses:**

(1) Missing qualitative comparisons with prior methods, limiting insights into this model’s advantages in style fidelity and layout accuracy.

(2) The contributions over previous approaches could be articulated more clearly, especially regarding the effectiveness of the layout-glyph separation.

(3) The organization could be refined for readability, as the methods section contains complex explanations that could benefit from clearer structuring.

**Questions:**

(1) Could more details be provided on how the layout-glyph separation specifically enhances performance in comparison to prior models?

(2) Would additional experiments on style consistency across diverse text lines clarify the benefits of this approach?

(3) Could this method can be adapted to non-Chinese scripts or connected handwriting styles?

---

> ### Author Response · Authors · 2024-11-14
>
> Thank you for the thoughtful and positive feedback!  We greatly appreciate for the recognition of the innovative aspects of our approach and the thorough evaluation of our model's effectiveness. In the following, we provide detailed responses to the reviewer’s queries one by one:
>
> ---
>
> >Comment1: Missing qualitative comparisons with prior methods to prove the advantages in style fidelity and layout accuracy.
>
> **Answer**:  This is a valuable suggestion.
> + For style fidelity: **we have supplemented** style qualitative comparison with existing methods, in appendix A.2.5.
> +  For layout accuracy: As stated in Section 4.3.1, previous work models the layout of text lines via Gaussian distribution in a model-free manner. We have reproduced this method and demonstrate the effectiveness and necessity of our layout generation approach in Sections 4.3&4.4. In particular, we visually demonstrate in Figure 7 that overly simplistic model-free methods fail to generate layouts with distinct personal characteristics, making them easily recognizable at a glance. **We have supplemented** more details in Section 4 of the supplementary materials.
>
> >Comment2 & Question 1:  The contributions over previous approaches could be articulated more clearly, especially regarding the effectiveness of the layout-glyph separation; Could more details be provided on how the layout-glyph separation specifically enhances performance in comparison to prior models?
>
> **Answer**:  Thank you for your suggestion! This is an important issue in our approach. I will provide more details about the layout-glyph separation:
>
> + Contribution: As far as I know, we are **the first to complete the generation of handwritten Chinese text lines**.  The significant contribution of the  layout-glyph separation, therefore, is its ability to effectively address this issue. In Chinese text lines, the structure and the position of each glyph  are both crucial. However, previous methods for single-character generation typically focus on transferring a standard character template style to handwritten style, which completely neglects modeling the positional relationships between multiple characters, making them **unsuitable for text line** generation tasks.
> + Motivation : When extending from single characters to text lines, a key challenge lies in how to represent the content information of the text line. In our extended experiments (Section 3 of the supplementary materials), we conducted end-to-end string generation experiments by concatenating the embeddings of individual characters to form the embedding of a text line. However, without further improvements, we found that the model was capable of generating short sequences but **encountered difficulties with longer sequences**. We attribute this issue to the complexity of training a single model to simultaneously learn both the structure of the characters and their relative positions. To reduce the learning complexity of the diffusion model, we decomposed the full probabilistic model and decoupled the layout component from the glyphs. While the performance is basically satisfactory, we make the training and sampling process simple and stable.
>
> >Comment3: The organization could be refined for readability.
>
> **Answer**:  Thank you for pointing out the writing issues. We will carefully revise and supplement the content based on your feedback to improve the readability and quality of our paper.
>
> >Question2:  Would additional experiments on style consistency across diverse text lines clarify the benefits of this approach?
>
> **Answer**: This suggestion is highly insightful! We have **improved the subjective experiments** in Section 4.4 considering this factor. Specifically, we now construct each test sample by combining a real handwritten text line from a particular author with **more than one** synthetic text lines. The participants are tasked with determining whether the lines were written by the same person. If there are inconsistencies in style between the synthetic sample and the real sample, or between different synthetic lines, the testers are likely to notice. We believe this improvement makes our experiment more complete.
>
> >Question3:  Could this method can be adapted to non-Chinese scripts or connected handwriting styles?
>
> **Answer**:
> + Our method is particularly suited for language systems where characters are relatively independent, such as Chinese, Japanese, Korean or even mathematic formula.
> + As mentioned in Section 5 of the original paper, the limitation of layout-glyph seperation lies in its difficulty in replicating connected handwriting styles. However, in Section 3 of the supplementary material, we have explored and made attempts at generating connectedstyle writing in an end2end manner.
>
> ---
> Thank you again for your valuable suggestions for improving our paper！I hope our response can effectively address your concerns.

---

> ### Author Response · Authors · 2024-11-17
>
> Dear Reviewer,
>
> We have just submitted the supplementary material. In response to your questions and suggestions, we have made the following revisions:
> + In Section 1, we present a visual comparison with state-of-the-art methods, which enhances the completeness of our paper.
> + In Section 2, we have improved the visual subjective evaluation experiments to not only assess the effectiveness of style imitation but also evaluate the consistency between different lines of text. Thank you for your valuable suggestions!
> + In Section 3, we demonstrate how the designed model can be extended to an end-to-end generation framework, addressing the potential issue of character linking in generated text. We also compare the advantages and disadvantages of the two methods at the current stage. This also demonstrates the motivation and benefits of decoupling the layout from the glyphs.
> + In Section 4, we present more visual comparison results of the layout generation methods with previous approaches and carefully elaborate on our advantages and contributions.
>
> We have also made these revisions to the corresponding sections of the original text as per the suggestions. Thank you again for your thoughtful feedback! If you have any additional questions or suggestions, please feel free to let us know!

---

> ### Author Response · Authors · 2024-11-25
> **Looking forward to your valuable feedback**
>
> Dear Reviewer,
>
> Thank you once again for your thoughtful and constructive feedback! During the rebuttal process, base on your suggestions, we have made the following improvements：1) We incorporated additional visual comparison experiments. 2) We provided a detailed explanation of the motivation of layout-glyph separation and its advantages over previous methods. 3) We improved the subjective evaluation by comparing the consistency of style across different generated samples, making the assessment more comprehensive. 4) Furthermore, we explored how the model  can be expanded for generating connected handwritings.
>
> We genuinely hope that our responses have effectively addressed your questions and concerns. If you have any further inquiries or need additional clarification, please feel free to reach out. We are eager to refine and enhance the contribution of our work. We sincerely hope to receive your approval. Once again, thank you for your valuable feedback!
>
> Sincerely,
>
> The authors.

---

> > ### Comment · Reviewer_kxE9 · 2024-11-25
> >
> > Thanks for the detailed responses, I have no further questions.

---

> > > ### Author Response · Authors · 2024-11-25
> > >
> > > We sincerely thank the reviewer for their positive response to our primary contribution and for increasing the score in recognition of our improvements.
> > >
> > > Warmly regards!

---

### Official Review · Reviewer_iC1s · 2024-11-04

**Soundness:** 4
**Presentation:** 3
**Contribution:** 3
**Rating:** 6
**Confidence:** 4

**Summary:**

The paper introduces a novel approach for generating online handwritten Chinese text with specific styles. The authors naturally divide a text line into two components: layout and glyphs, and design a text line layout generator coupled with a diffusion-based stylized font synthesizer to address this challenge hierarchically. The layout generator autoregressively generates the positions for each glyph based on text content and provided style references, while the font synthesizer generates each font at its position while imitating the calligraphy style extracted from the given style references. Experiments on the CASIAOLHWDB demonstrate the method's capability to generate structurally correct and indistinguishable imitation samples.

**Strengths:**

1. The study proposes a hierarchical method to address the under-explored task of online handwritten Chinese text line generation.
2. By decoupling layout generation from glyph generation, the method offers more flexibility in handling the generation of text lines, which is particularly useful when dealing with complex Chinese characters.
3.  The experiments conducted on the CASIA-OLHWDB database indicate high performance in imitation sample generation, demonstrating the effectiveness of the method.

**Weaknesses:**

1. While decoupling layout and glyph generation increases flexibility, it may also add to the model's complexity, potentially affecting training and inference efficiency.
2. Are there any application scenarios for this task? The author could analyze its practicality.
3. The paper mentions difficulties in imitating styles with extensive cursive connections between characters due to the independent generation of each character, indicating potential limitations in handling certain calligraphic styles.

**Questions:**

Please see Weaknesses

---

> ### Author Response · Authors · 2024-11-14
>
> We are grateful for the positive feedback and valuable questions!  We are truly grateful that the reviewer acknowledges the novelty, clarity, and flexibility of our proposed method. In the following, we provide detailed responses to the reviewer’s queries one by one:
>
> ---
>
> >Comment1:  Consideration about the model's complexity, training and inference efficiency.
>
> **Answer**:  This is a very important consideration, in this response, we will first clarify more about the motivation behind decoupling and then provide a detailed analysis of the efficiency aspects, particularly in relation to training and sampling.
> + Motivation: We actually have also experimented based on our 1D U-Net generator to generate strings end-to-end. However, without further improvements, we found that the model can generate short sequences (e.g., two to five consecutive characters) but encounters difficulties with longer sequences. We attribute this issue to the complexity of having a single diffusion model simultaneously learn both the structure of the characters and their relative positions. To **reduce the learning complexity** of the diffusion model, we decompose the full probabilistic model and decouple the layout component from glyphs.
> + Training and inference efficiency: 1) During the training phase, as described, the diffusion model only needs to learn the generation of individual characters, while the layout generator is responsible for planning the size and position of each character. This effectively reduces the complexity of training diffusion model.   2) In the sampling phase, since the generation processes of the two components are decoupled, as shown in Figure 3, **they can be fully parallelized**. The character generator can simultaneously generate all characters in a string within a batch, while the layout generator has already planned their positions. Additionally, as detailed in the appendix, the task is not computationally intensive, therfore our layout generator is lightweight (a 2-layer LSTM with a hidden size of 128).
>
> >Comment2:  Analyze the application scenarios for this task.
>
> **Answer**:  We currently have three considerations for potential application directions:
> + We can provide **personalized writing services**. As is widely known, Chinese comprises thousands of commonly used characters. Users can submit a short passage (e.g., a dozen characters) of any content as a stylistic reference sample. We can then mimic their writing habits to generate handwritten samples of any length and content.
> + It can be used **in the field of education**. Since the online data we generate includes dynamic information about the writing process, it can teach the stroke structure and order of different Chinese characters.
> + Used for **data augmentation**: In today's Chinese text recognition field, the training data available is extremely limited (commonly only CASIA OLHWDB). We have made initial progress in this area. We use all real-world data (about 60,000 text lines) as training data, train the text line recognizer using 3-layer LSTM with CTC Loss, achieving 85.5 AR 85.7 CR on the Icdar2013 test set. The results of data augmentation are as follows, demonstrating the application value of our method in this task.
>
> |            |  AR  |  CR  |
> |:----------:|:----:|:----:|
> | Real data  | 85.5 | 85.7 |
> | Real + 2.5W aug | 88.3 | 88.7 |
> | Real + 5.0W aug | 90.9 | 91.2 |
>
> One advantage of our approach is that the generated data inherently includes strong positional labels for characters, which makes it suitable for data augmentation in recognition tasks that **require strongly labeled data** for training, such as character segmentation.
>
>
> >Comment3:  Limitations in handling certain calligraphic styles.
>
> **Answer**:  The hierarchical generation method do have a weakness in not being able to generate inter-character connections. A potential solution is the end-to-end framework or to make a trade-off by training an end-to-end generation model for a limited number of characters, and then combining it with the layout generation model. We present some possibilities for extending to end-to-end framework in Section 3 of the supplementary material. We will strive to make up for the shortcomings of our methods in our future work.
>
> ---
> Hope my response effectively addresses your concerns, if you have any remaining questions or need further information about our study, please feel free to let us know!

---

> ### Author Response · Authors · 2024-11-17
>
> Dear Reviewer,
>
> We have just submitted the supplementary material. In response to your questions and suggestions:
> + In Section 3, we demonstrate the potential of **extending our designed 1D-UNet** model to an end-to-end generation paradigm, thereby addressing the issue of generating connected strokes between characters.
>
> We have also made revisions to the corresponding sections of the original text as per the suggestions. Thank you for your recognition of our work! If you have any further questions or suggestions, we would be honored to engage with you！

---

> ### Author Response · Authors · 2024-11-25
> **Looking forward to your valuable feedback**
>
> Dear Reviewer,
>
> Thank you once again for your thoughtful and constructive feedback! During the rebuttal process, base on your suggestions, we have made the following improvements: 1) We clarified the complexity and efficiency of the method; 2) We provided an analysis of the application scenarios and have demonstrated its effectiveness in data augmentation; 3) We investigated ways to expand the model for generating continuous handwriting.
>
> We sincerely hope that our responses have satisfactorily addressed your questions and concerns. If you have any further inquiries or require additional clarification, please do not hesitate to let us know. We are more than happy to refine and strengthen the contribution of our work. Once again, thank you for your invaluable input !
>
> Sincerely,
>
> The authors.

---

### Author Response · Authors · 2024-12-04
**The summary of rebuttal**

Dear AC and reviewers,

In this section, we aim to highlight the contributions of our work and summarize the key points from our rebuttal. Our study proposes a hierarchical approach to solve the challenging task of handwritten Chinese text line generation, which has been rarely explored. We introduce a in-context layout generator based on the next-token-prediction paradigm and construct a 1D-UNet-based diffusion denoiser for font generation. Our method can also be applied to other complex structured handwritten data and is suitable for use as a data augmentation technique that generates data with strong labeling information.

During the rebuttal stage, we are pleased that the reviewers have acknowledged the novelty and contribution of our methods. We provided detailed responses to the reviewers' questions regarding the method's specifics. Following their suggestions, we not only improved our manuscript, but also added quantitative and qualitative experiments in the appendix and supplementary materials to further validate the effectiveness and scalability of the method. We believe that these revisions adequately address all the reviewers' concerns and further strengthen the contributions of our study.

We sincerely appreciate the time and effort that the reviewers have dedicated to evaluating our manuscript. Their insightful comments and constructive feedback have significantly improved the quality of our work. We are also grateful for the AC's efforts throughout the rebuttal process.

Warm regards,

The Authors

---

### Meta-Review · Area_Chair_6SAB · 2024-12-20

**Metareview:**

This paper studies online handwritten Chinese text generation. After rebuttal, the overall rating of this paper is above the marginal acceptance threshold, but with mixed scores of 5,6,6,8. The Area Chair has read the paper, all reviews, and the authors' rebuttal. The main strengths of the paper include 1) the task of the online handwritten Chinese text is considered to be important and the proposed method is well motivated, 2) the proposed decoupling of layout and glyph is reasonable and innovative, 3) the experimental design is thorough, and the results presented are convincing.

During the rebuttal phase, the authors addressed most of the reviewers' concerns, leading Reviewers iC1s, Up7T, and kxE9 to lean toward accepting the paper. However, Reviewer BU7e remained concerned about the novelty of the proposed multi-scale style encoder with contrastive loss, maintaining a rating of 5. Despite this, considering all review comments and the paper's other contributions, the AC agrees that the paper meets the acceptance threshold. Additionally, during the discussion phase, Reviewer BU7e also agrees that the paper could be accepted despite the concerns regarding its novelty.

The recommendation is acceptance. The authors are advised to carefully revise the manuscript by incorporating the reviewers' suggestions to enhance its quality further.

**Additional Comments On Reviewer Discussion:**

After rebuttal, Reviewers iC1s, Up7T, and kxE9 lean toward accepting the paper. Reviewer BU7e also agrees that the paper could be accepted despite the concerns regarding its novelty.

---

### Decision · Program_Chairs · 2025-01-22

Accept (Poster)